# Catalytic ozonation mechanism over M₁-N₃C₁ active sites

**Dingren Ma** [1], **Qiyu Lian**[1], **Yexing Zhang**[1], **Yajing Huang**[1], **Xinyi Guan**[1], **Qiwen Liang**[1], **Chun He**[1], **Dehua Xia** [1,3] ✉, **Shengwei Liu** [1,3] ✉ & **Jiaguo Yu** [2,3] ✉

The structure-activity relationship in catalytic ozonation remains unclear, hindering the understanding of activity origins. Here, we report activity trends in catalytic ozonation using a series of single-atom catalysts with well-defined M₁-N₃C₁ (M: manganese, ferrum, cobalt, and nickel) active sites. The M₁-N₃C₁ units induce locally polarized M − C bonds to capture ozone molecules onto M atoms and serve as electron shuttles for catalytic ozonation, exhibiting excellent catalytic activities (at least 527 times higher than commercial manganese dioxide). The combined in situ characterization and theoretical calculations reveal single metal atom-dependent catalytic activity, with surface atomic oxygen reactivity identified as a descriptor for the structure-activity relationship in catalytic ozonation. Additionally, the dissociation barrier of surface peroxide species is proposed as a descriptor for the structure-activity relationship in ozone decomposition. These findings provide guidelines for designing high-performance catalytic ozonation catalysts and enhance the atomic-level mechanistic understanding of the integral control of ozone and methyl mercaptan.

Severe warm-season ozone ($O_3$) pollution (>200 μg m$^{-3}$), mainly generated from the photochemical reactions between nitrogen dioxides and volatile organic compounds (VOCs) in the air under sunlight, is occurring with increasing frequency on a global scale[1-3]. For the health risks of long-term $O_3$ exposure, the World Health Organization has set a new indicator that the peak value of warm-season $O_3$ needs to be less than 60 μg m$^{-3}$ [4]. Catalytic decomposition of $O_3$ into oxygen ($O_2$) has recently become a popular technology for controlling $O_3$ pollution[5-7]. However, as the precursor of $O_3$, the degradation of VOCs is often neglected, which largely hinders the efforts of $O_3$ decomposition technology. In particular, sulfur-containing volatile organic compounds (SVOCs, e.g., hydrogen sulfide ($H_2S$), methyl mercaptan ($CH_3SH$), etc.) can not only generate $O_3$ but also lead to a direct threat to the human nervous system and environmental safety even at an extremely low concentration[8,9]. The simultaneous removal of SVOCs

and $O_3$ from the air is thus of high importance. Catalytic ozonation for SVOCs degradation at room temperature has been regarded as an ideal technology to simultaneously control both pollutants with attractive advantages of safety, high efficiency, and environmental friendliness[10-12]. Although numerous catalysts have been designed and applied in catalytic ozonation, the major perspectives are focusing on adsorption-activity or reactive oxygen species-activity relationships[13,14]. Few studies concerned the structure-activity relationships involved in catalytic ozonation, which could obscure the origin of the activity and limit the optimal design of the catalysts. Therefore, it is imperative to create a descriptor that can well express the structure-activity relationships in catalytic ozonation.

Catalytic ozonation is a surface science, in which surface chemical reactions occurring on active catalytic sites after $O_3$ adsorption directly determine the mechanism-dependent activity and selectivity[15].

¹School of Environmental Science and Engineering, Guangdong Provincial Key Laboratory of Environmental Pollution Control and Remediation Technology, Sun Yat-sen University, Guangzhou 510275, China. ²Laboratory of Solar Fuel, Faculty of Materials Science and Chemistry, China University of Geosciences, 68 Jincheng Street, Wuhan 430078, China. ³These authors jointly supervised this work: Dehua Xia, Shengwei Liu, Jiaguo Yu. ✉e-mail: xiadehua3@mail.sysu.edu.cn; liushw6@mail.sysu.edu.cn; yujiaguo93@cug.edu.cn

Oyama et al. revealed that the $O_3$ decomposition reactions include the dissociation of an adsorbed $O_3$ to form a gas-phase $O_2$ and a surface atomic oxygen (*O), the reaction of *O with a gas-phase $O_3$ to form a gas-phase $O_2$ and a surface peroxide species (*$O_2$), the decomposition of *$O_2$ to form a gas-phase $O_2$[16]. Recent studies have proposed that in catalytic ozonation reactions, the *O and *$O_2$ not only directly participate in oxidation reactions but also regulate the generation of other reactive oxygen species, such as hydroxyl radicals (•OH), singlet oxygen ($^1O_2$), and superoxide radicals (•$O_2^-$), by reacting with $H_2O$, thus indirectly affecting catalytic activity[17,18]. It is reasonable to consider that *O/*$O_2$ can be the potential descriptors to directly reflect the structure-activity relationships in catalytic ozonation. Nevertheless, an in-depth understanding of the surface chemical reactions is limited by the unclear active sites[19]. Therefore, it is necessary to investigate the structural and electronic states of *O/*$O_2$ on the well-defined active sites to describe the structure-activity relationship in catalytic ozonation.

Single-atom catalysts (SACs) with well-defined, single, and homogeneously distributed active centers provide a promising platform for investigating the structure-activity relationship[20,21]. In SACs, the isolated metal (M) atoms are generally anchored onto carbon nanomaterials by coordinating with the nitrogen dopants, forming the $M_1$-$N_4$ coordination[22,23]. Nevertheless, the strong electronegativity of the $D_{4h}$ (square plane) symmetric N atoms in the $M_1$-$N_4$ coordination can undesirably alter the electronic states of single metal atoms, thus increasing the adsorptive free energy towards reaction intermediates, which mainly limits the formation of reactive oxygen species that can directly determine the mechanism-dependent activity and selectivity[20,23,24]. To tackle this problem, secondary heteroatomic dopants (S and P) with relatively weak electronegativity are usually used to optimize the electronic properties of the active metal centers[25,26]. Recently, a simple temperature-tuned N-coordination strategy to manipulate single-atom $M_1$-$N_3C_1$ coordination was reported[27]. It is predicted that the C atoms with relatively weak electronegativity in the $M_1$-$N_3C_1$ coordination induce electron aggregation on the M atoms, thus causing higher electron density states to correctly capture the electrophilic $O_3$ molecules onto the M atoms[28,29]. Further considering that the unfilled 3d orbitals can enhance charge transfer between the M atoms and *O/*$O_2$, oxophilic transition metals (Mn, Fe, Co, and Ni) are selected as the M centers[30]. The M centers, with different 3d orbital electrons acting as the electron shuttles, regulate the structural and electronic states of *O/*$O_2$, as well as the mechanisms and kinetics for both the nonradical (*O, *$O_2$, and $^1O_2$) and radical (•$O_2^-$ and •OH) pathways, thus controlling the catalytic ozonation performance[14,31].

In this study, we systematically investigate the intrinsic effects of the electronic properties of $M_1$-$N_3C_1$ active sites (M: Mn, Fe, Co, and Ni) on $O_3$ decomposition and catalytic ozonation. Aberration-corrected high-angle annular dark-field scanning transmission electron microscope (AC HAADF-STEM), X-ray absorption fine structure (XAFS), and X-ray photoelectron spectroscopy (XPS) reveal the well-defined $M_1$-$N_3C_1$ coordination. The as-prepared SACs exhibit excellent $O_3$ decomposition and catalytic ozonation performance, where the performance and product distribution are controlled by the single metal atoms. Detailed in situ characterization combined with density functional theory (DFT) calculations unravels the structure-activity relationship in the $O_3$ decomposition (the desorption free energy of *$O_2$) and catalytic ozonation (the reactivity of *O) on the $M_1$-$N_3C_1$ active sites.

## Results

### Synthesis and structure characterization

Supplementary Fig. 1 depicts the basic procedures to prepare the MNC (M: Mn, Fe, Co, and Ni) SACs. The single metal atoms are evenly distributed on the nitrogen-doped carbon substrate after the pyrolysis

and carbonization under $N_2$ flow. Notably, the $M_1$-$N_3C_1$ coordination can be easily formed attributed to the heavy vanishing of nitrogen species at a high reaction temperature of 800 °C[27].

The morphologies of the MNC SACs were studied by scanning electron microscopy (SEM) and transmission electron microscopy (TEM). As shown in Supplementary Fig. 2, the samples all exhibit the nanosheet-stacking structure, which is consistent with the type-IV $N_2$ adsorption-desorption isotherms with an H3 hysteresis loop (Supplementary Fig. 3)[32]. The nanosheet-stacking structure results in a high specific surface area (-200 $m^2$ $g^{-1}$, Supplementary Table 1), which enables the exposure of abundant active sites and efficient mass transportation[33]. The TEM results further confirm the nanosheet-stacking structure and the absence of metal nanoparticles (Supplementary Fig. 4), while bright spots in the atomic range are detected in the samples via the aberration-corrected high-angle annular dark-field scanning transmission electron microscopy (AC HAADF-STEM), highlighting the presence of atomically dispersed metal atoms (Fig. 1a–d). The STEM-coupled energy-dispersive spectroscopy (EDS) element mapping shown in Fig. 1e corroborates the presence of C, N, and Co elements, and the homogeneous distribution of Co atoms throughout the CoNC[14]. Moreover, the control carbon material (NC) also exhibits the nanosheet-stacking structure (Supplementary Fig. 5), indicating that the introduction of single metal atoms does not alter the morphology of the matrices.

The X-ray diffraction (XRD) patterns show two broad diffraction peaks around 26° (002, graphite) and 44° (101, graphite) in Fig. 2a, which are attributed to the disordered and defective carbon structures[34]. Notably, there are no diffraction peaks related to crystalline metals or metal oxides, as further evidenced by the Raman spectra (Supplementary Fig. 6). The chemical elements and electronic states of the MNC investigated by X-ray photoelectron spectroscopy (XPS, Supplementary Figs. 7–10) show that the metal species only present in their oxidation states ($Mn^{2+}$ 641.53 eV, $Fe^{2+}$ 709.96 eV, $Co^{2+}$ 780.62 eV, $Ni^{2+}$ 854.72 eV) rather than metallic states[30,35–37]. Moreover, there are multiple N dopants in the catalysts, including pyridinic-N (398.02 eV), metal-N (399.15 eV), pyrrolic-N (400.00 eV), graphitic-N (400.61 eV), and oxidized N (402.80 eV)[22,38]. This is further confirmed by the Fourier transform infrared (FTIR) spectra, as the two peaks at 1264 and 1535 $cm^{-1}$ assigned to the stretching vibrations of C − N and C = N, respectively, are detected (Supplementary Fig. 11)[39]. The surface elemental contents were calculated and summarized in Supplementary Table 2. The samples exhibit similar metal contents, which are confirmed by inductively coupled plasma optical emission spectrometry (ICP-OES, MnNC 0.99 wt%, FeNC 1.04 wt%, CoNC 1.00 wt%, and NiNC 1.00 wt%) in Supplementary Table 1. Interestingly, the metal-N content is around 3 times higher than the metal content, which may reveal the $M_1$-$N_3C_1$ coordination formed in the MNC. It is worth noting that the matrix structure of the MNC SACs remains consistent, except for variations in the doping type of single metal atoms. This fact is further supported by structural characterization results (Fig. 2a, Supplementary Fig. 6, and Supplementary Fig. 11), which show no discernible differences between the NC and the MNC SACs.

The $M_1$-$N_3C_1$ coordination was investigated and confirmed by X-ray absorption fine structure (XAFS) analysis (CoNC as representative). One noticeable peak around 1.5 Å in the extended-XAFS (EXAFS) profile of CoNC (Fig. 2b), corresponding to the Co−N/C bond, is consistent with the characteristics of an isolated Co atom surrounded by nitrogen/carbon atoms[20,40]. Moreover, no characteristic peaks for Co−Co contribution at 2.1 Å can be found, further indicating the presence of atomically dispersed Co atoms[38]. These results are further confirmed by the wavelet transform (WT) of EXAFS (Supplementary Fig. 12). In the X-ray absorption near-edge structure (XANES) profiles (Fig. 2c), the noticeable intensity of the pre-edge peak (1s-to-3d, 7710 eV) of CoNC reveals the asymmetric Co atom coordination, which is significantly different from the negligible peak caused by the $D_{4h}$

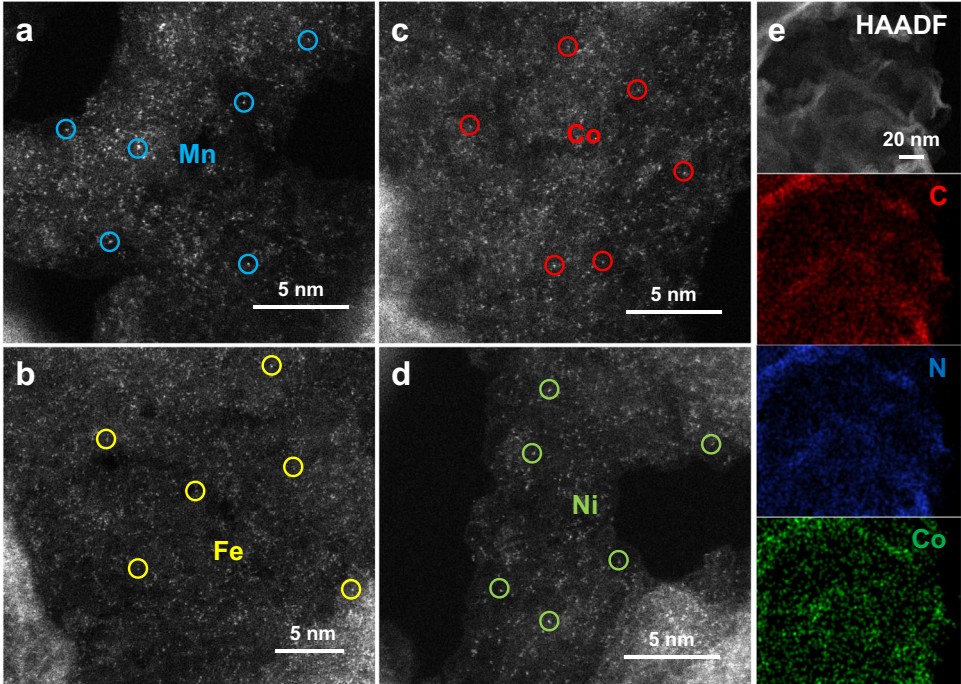

**Fig. 1 | Structural characterization results. a–d** Aberration-corrected high-angle annular dark-field scanning transmission electron microscope (AC HAADF-STEM) images of the MnNC (**a**), FeNC (**b**), CoNC (**c**), and NiNC (**d**). **e** Elemental mapping of the CoNC.

(square planar) symmetry of the Co-N$_4$ coordination in CoPc[23]. Moreover, the valence state of Co atoms in CoNC is lower than that in CoPc since the near-edge absorption energy of CoNC is located between Co foil and CoPc[21]. Combined with the XPS results, it is speculated that a C atom with relatively weak electronegativity replaces a N atom to form the Co$_1$-N$_3$C$_1$ coordination. Notably, the EXAFS fitting (Fig. 2d and Supplementary Table 3) confirms that the Co$_1$-N$_3$C$_1$ coordination dominates the CoNC catalyst. The overall structural analyses indicate that the MNC SACs have the well-defined M$_1$-N$_3$C$_1$ units, which are anchored onto the same nitrogen-doped carbon matrices.

The theoretical models of the MNC SACs were constructed based on the XAFS analyses, as shown in Supplementary Fig. 13. Moreover, the corresponding M$_1$-N$_4$ theoretical models were established in Supplementary Fig. 14. The calculated projected density of states (PDOS) results in Supplementary Fig. 15 show that the metal 3$d$ orbitals exhibit significant orbital electronic coupling to the C 2$p$ and N 2$p$ orbitals, respectively, confirming the stability of the M$_1$-N$_3$C$_1$ coordination[41]. The deformation charge density and the Bader charge were performed to study the precise electron distribution[13]. The redistribution of electrons is observed in the M$_1$-N$_3$C$_1$ and M$_1$-N$_4$ units (Fig. 2e, f and Supplementary Figs. 16, 17), with the loss and gain of electrons at the metallic and nonmetallic sites, respectively. Interestingly, the electronic states of the M$_1$-N$_3$C$_1$ units (Co −0.75e$^-$) significantly differ from the M$_1$-N$_4$ units (Co −0.79e$^-$), which is consistent with the results of XAFS. Specifically, the weaker electronegativity of the C atom than the N atom in the M$_1$-N$_3$C$_1$ coordination induces the locally polarized M − C bond, forming the high electron density regions. Each functional M$_1$-N$_3$C$_1$ region with high electron density can correctly capture the electrophilic O$_3$ molecules and work as an electron shuttle to the catalytic ozonation, thus leading to excellent catalytic performance[13]. More importantly, different electronic states are observed among metal centers sharing the same N$_3$C$_1$ coordination environment (Supplementary Fig. 16), which can be attributed to the different energy states of the 3$d$ orbital electrons and orbital interactions between single metal atoms and C/N atoms (Supplementary Fig. 15). Therefore, the single metal atoms govern the electronic states of the M$_1$-N$_3$C$_1$ units

(Supplementary Fig. 18), forming different $d$-band centers (MnNC (−0.18 eV) > FeNC (−0.34 eV) > CoNC (−0.66 eV) > NiNC (−1.73 eV)), which will further determine the structural and electronic states of surface reaction intermediates as well as the reaction mechanisms and kinetics for the catalytic ozonation[42].

## Ozone decomposition and catalytic ozonation performance

The performance of O$_3$ decomposition and catalytic ozonation for CH$_3$SH degradation was evaluated on the well-defined M$_1$-N$_3$C$_1$ active sites. The performance of O$_3$ dynamic decomposition in Fig. 3a shows that all the MNC catalysts achieve complete O$_3$ decomposition. Notably, the O$_3$ decomposition efficiencies after 60 min are clearly influenced by the anchored single metal atoms in the order of NiNC (100.0%) > CoNC (99.7%) > FeNC (99.4%) > MnNC (98.6%) > NC (97.4%). The decrease in O$_3$ decomposition activities can be attributed to the occupation of active sites by intermediates, while the O$_3$ decomposition performance can be restored after 1 h of treatment at 100 °C under the N$_2$ atmosphere (Supplementary Fig. 19)[6]. Moreover, the metal-dependent performance of O$_3$ decomposition can be attributed to variances in the interactions between single metal atoms and intermediates, which will be discussed later in the DFT calculations.

The efficient O$_3$ decomposition will generate abundant reactive oxygen species that are capable of the oxidative degradation of CH$_3$SH[43]. The catalytic ozonation for CH$_3$SH degradation also shows a metal-dependent profile in Fig. 3b. The CoNC catalyst performs the most efficient degradation of CH$_3$SH (100%), significantly higher than the others (FeNC 87.3%, NiNC 81.6%, MnNC 73.1%) at the same condition. In contrast, the NC shows the lowest catalytic activity (58.5%), which further decreases to 43.2% after 60 min of reaction. Therefore, it can be inferred that the M$_1$-N$_3$C$_1$ active sites play a crucial role in catalytic ozonation reactions. The organic compounds in the exhaust gas (Fig. 3c–f and Supplementary Table 4) were identified by proton transfer reaction time-of-flight mass spectrometry (PTR-TOF-MS). In this condition, methanol (CH$_4$O, 33), dimethyl sulfoxide (C$_2$H$_6$OS, 79), and dimethyl sulfone (C$_2$H$_6$O$_2$S, 95) are the main byproducts. It is important to point out that the concentration of these byproducts is

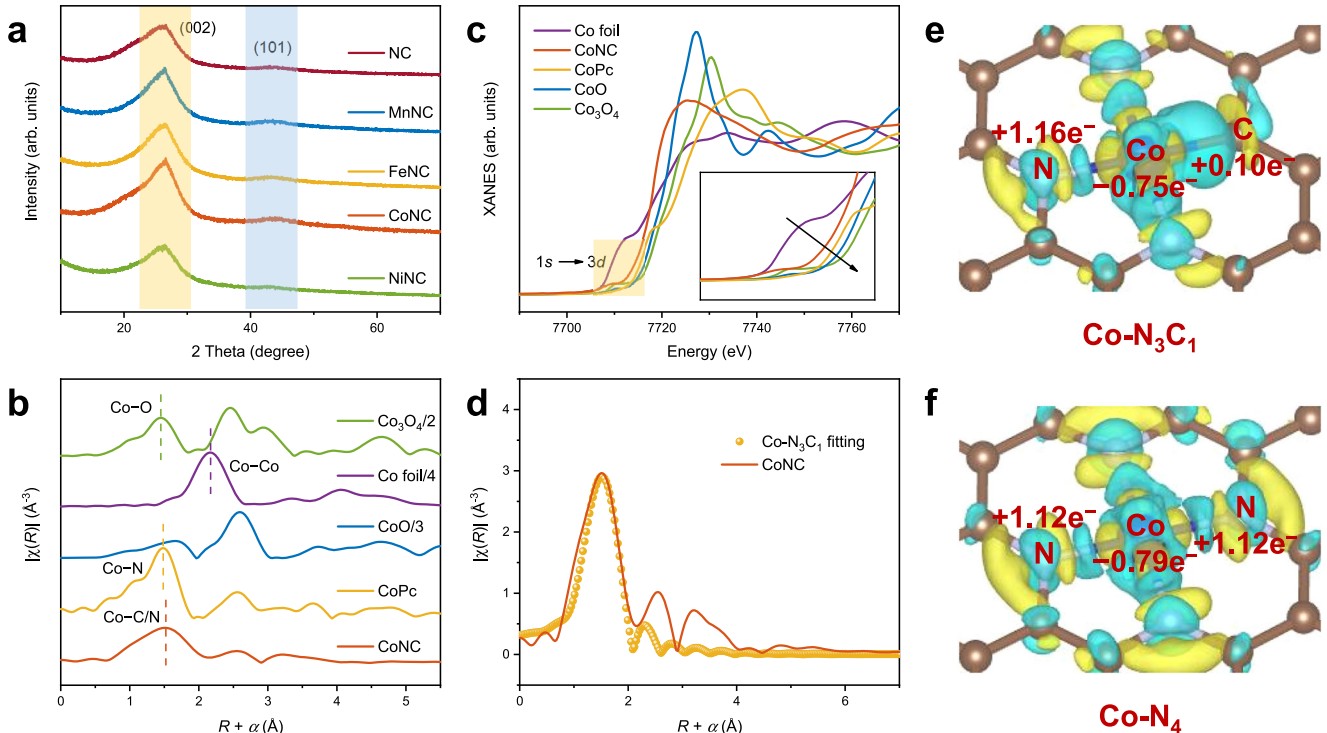

**Fig. 2 | Atomic structure analysis. a** Powder X-ray diffraction (XRD) patterns of the NC and MNC catalysts. **b** Co K-edge X-ray absorption near-edge structure (XANES) spectra of the CoNC and reference materials. **c** The extended X-ray absorption fine structure (EXAFS) spectra of the CoNC and reference materials. **d** Corresponding extended X-ray absorption fine structure (EXAFS) fitting curves of the CoNC at R space. **e**, **f** Charge difference isosurfaces and Bader charge of the Co-$N_3C_1$ (**e**) and Co-$N_4$ (**f**). Isosurfaces level = 0.005. All lengths are given in Å. The blue, brown, and silver balls denote Co, C, and N atoms, respectively. The blue and yellow isosurfaces represent charge accumulation and depletion in the space, respectively.

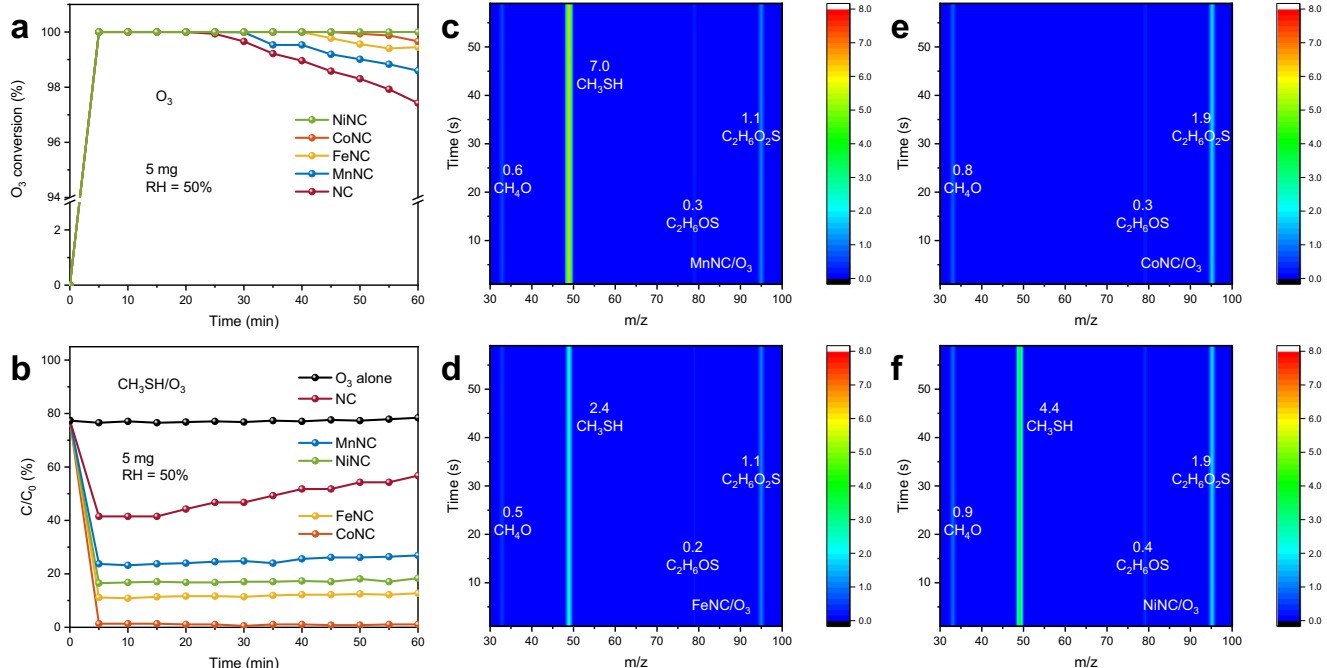

**Fig. 3 | Ozone decomposition and catalytic ozonation performance. a** Ozone ($O_3$) dynamic decomposition tests over the NC and MNC catalysts. **b** Catalytic ozonation for methyl mercaptan ($CH_3SH$) degradation tests over the NC and MNC catalysts. **c**–**f** The concentrations of $CH_3SH$ and typical intermediates in the outlet gases of the MNC catalysts (**c**–**f** MnNC, FeNC, CoNC, and NiNC) after the catalytic ozonation reactions for 60 min determined by proton transfer reaction time-of-flight mass spectrometry (PTR-TOF-MS).

inversely proportional to the catalytic activity (Supplementary Fig. 20), indicating that the $CH_3SH$ mineralization efficiency based on the oxidation of reactive oxygen species also shows the same metal-dependent profile (CoNC > FeNC > NiNC > MnNC). Therefore, the metal-dependent catalytic ozonation performance relies on the reactive oxygen species generated on the $M_1$-$N_3C_1$ active sites, which will be discussed later in the DFT calculations. Notably, all the catalysts maintain the 100% decomposition of $O_3$ (Supplementary Fig. 21) within 60 min, which indicates that the reactions between $CH_3SH$ and the reactive oxygen species promote $O_3$ decomposition, thus achieving simultaneous degradation of $O_3$ and $CH_3SH$. In particular, the CoNC catalyst performs the complete degradation of $O_3$ and $CH_3SH$ and the harmless emission of exhaust gas. By integrating the performance results of the $O_3$ decomposition and catalytic ozonation, it can be inferred that the $M_1$-$N_3C_1$ units in the MNC catalysts play a pivotal yet distinct role in these two reactions.

Interestingly, the degradation efficiencies of $CH_3SH$ by the MNC catalysts in the air (Supplementary Fig. 22) show a metal-dependent profile similar to that observed under the $O_3$ conditions. The order of $CH_3SH$ degradation efficiencies achieved is CoNC (84.5%) > FeNC (71.0%) > NiNC (41.5%) > MnNC (37.7%) > NC (35.8%). However, the $CH_3SH$ degradation efficiencies decrease to 73.0%, 61.4%, 30.7%, 19.4%, and 0% after 60 min, respectively, revealing the weak oxidation capacity of the MNC catalysts in the air. This conclusion is confirmed by the abundant byproducts present in the exhaust gas (Supplementary Figs. 23, 24) and the low percentage of $SO_x$ species ($SO_4^{2-}$/$SO_3^{2-}$) on the catalyst surface (Supplementary Fig. 25 and Supplementary Table 5). Therefore, the reactive oxygen species with high oxidation potential generated by $O_3$ decomposition play a key role in the oxidation of $CH_3SH$.

For practical applications, the catalytic ozonation performance of the MNC catalysts was investigated by varying the various variables, such as humidity, dosages, and contact time. When the relative humidity (RH) > 90% (Supplementary Fig. 26a), the $CH_3SH$ degradation efficiencies of CoNC, FeNC, NiNC, and MnNC increase to 100.0, 94.6, 89.0, and 82.8%, respectively. While, for the dry inlet gas (RH < 1%), the catalytic activities of the samples become different (Supplementary Fig. 26b). For CoNC, FeNC, NiNC, and MnNC, their $CH_3SH$ degradation efficiencies are 100.0%, 87.9%, 84.0%, and 78.3%, respectively, which subsequently decrease to 95.8, 74.0, 70.9, and 57.8% after 60 min. These results are consistent with the $O_3$ decomposition performance of the MNC catalysts under both dry and wet conditions. As shown in Supplementary Fig. 27a, under dry conditions, the MNC catalysts exhibit gradual deactivation after 20 min, despite achieving the complete removal of $O_3$. The stability of the MNC catalysts exhibits a gradual increase with an elevation in humidity levels (Fig. 3a and Supplementary Fig. 27b). When the humidity exceeds 90% (Supplementary Fig. 27b), the NiNC and CoNC remain active, while the FeNC and MnNC exhibit deactivation after 50 min and 35 min, respectively. These findings suggest that the $H_2O$ molecule not only promotes $O_3$ decomposition, but also enhances the catalytic ozonation performance of the MNC catalysts by facilitating $O_3$ decomposition. Notably, the enhanced decomposition of $O_3$ by $H_2O$ can be attributed to the reaction between $H_2O$ and the intermediates resulting from $O_3$ decomposition. This will be discussed later in DFT calculations. Apparently, increasing the catalyst dosage will effectively promote the degradation of $CH_3SH$ (Supplementary Fig. 28). In particular, 20 mg of the MNC catalysts all achieve the complete degradation of $CH_3SH$. Mass activity is an important metric to evaluate the catalytic activities of different catalysts, which can be given as the $CH_3SH$ degradation amount normalized to the active metal loading[44]. According to Supplementary Fig. 29, the mass activities of all the samples are higher than 700 ppm mg$^{-1}$, while the mass activity of CoNC reaches a staggering 1000 ppm mg$^{-1}$, which is 714 times higher than the commercial $MnO_2$ (Supplementary Fig. 30). Regarding the stability of the catalysts,

the CoNC catalyst still retains the 97.1% $CH_3SH$ degradation efficiency after the continuous running of 1000 min in Supplementary Fig. 31, which is better than the well-reported catalysts[13,14,43]. Moreover, the morphology and single Co atoms of the CoNC catalyst (Supplementary Figs. 32, 33) remain unchanged after the reaction, confirming the exceptional stability of the CoNC catalyst. Therefore, the MNC catalysts provide promising opportunities for the synergistic control of $O_3$ and $CH_3SH$.

## Identification of surface reaction intermediates

Combined with the above-mentioned analysis, the catalytic ozonation performance of the MNC SACs mainly relies on the reactive oxygen species and, more deeply, on the interactions of surface reaction intermediates and active sites[15,27]. Thus, in situ Raman was carried out to monitor and clarify the surface reaction intermediates and pathways of $O_3$ conversion on the $M_1$-$N_3C_1$ active sites. With continuous $O_3$ flow (Fig. 4a and Supplementary Fig. 34, after $N_2$ purging), new peaks are observed at 821 and 909 cm$^{-1}$, which are attributed to the surface peroxide species (*$O_2$, 1.35 V) and surface atomic oxygen (*O, 2.43 V), respectively, revealing that new intermediates are generated on the $M_1$-$N_3C_1$ active sites during $O_3$ decomposition[45,46]. Combined with the reported literature, the decomposition reaction sequence following $O_3$ adsorption on the $M_1$-$N_3C_1$ active sites includes the dissociation of $O_3$ to form a gas-phase $O_2$ and an *O, the reaction of *O with a gas-phase $O_3$ to form a gas-phase $O_2$ and an *$O_2$, the decomposition of *$O_2$ to form a gas-phase $O_2$[16,47]. The $O_3$ conversion pathway on the $M_1$-$N_3C_1$ active sites can be presented by the following equations (Eqs. 1–3).

$$M + O_3 \rightarrow O_2 + M - {}^*O \quad (1)$$

$$M - {}^*O + O_3 \rightarrow O_2 + M - {}^*O_2 \quad (2)$$

$$M - {}^*O_2 \rightarrow M + O_2 \quad (3)$$

Notably, the characteristic peak for *$O_2$ contribution can be found in the Raman spectra of the catalysts (Supplementary Fig. 35) performed in the air[14]. Therefore, the mild oxidation of $CH_3SH$ in the air (Supplementary Figs. 22–24) can be attributed to the presence of *$O_2$ (1.35 V)[14]. The $O_2$ activation pathway on the $M_1$-$N_3C_1$ active sites can be presented by the following equation (Eq. 4).

$$M + O_2 \rightarrow M - {}^*O_2 \quad (4)$$

The reactivity of *O/*$O_2$, directly determined by the $M_1$-$N_3C_1$ active sites, was investigated by the amperometric i-t curve (i-t) tests. As shown in Fig. 4b, the sequential addition of saturated $O_3$ solution and sodium thiomethoxide ($CH_3NaS$) solution shows a significant current impulse, indicating the formation of $M - {}^*O$ and/or $M - {}^*O_2$ complexes and the subsequent oxidation of $CH_3NaS$[48]. Apparently, the reactivity of the complexes ($M - {}^*O/{}^*O_2$, based on the degree of the current impulse in Supplementary Fig. 36) relies on the variety of single metal atoms following the order of CoNC > FeNC > NiNC > MnNC, which is consistent with the catalytic ozonation performance (Fig. 3b). This result confirms the direct involvement of $M - {}^*O/{}^*O_2$ in the catalytic ozonation reactions and highlights the direct effects of $M - {}^*O/{}^*O_2$ reactivity on catalytic ozonation performance. Besides being directly involved in oxidation reactions, the *O/*$O_2$ also regulates the generation of other reactive oxygen species (•OH, $^1O_2$, and •$O_2^-$) by reacting with $H_2O$[46]. The presence of other reactive oxygen species was studied by the electron spin resonance (ESR) tests[7]. When dimethylpyridine N-oxide (DMPO) is used as a trapping agent in Fig. 4c, the signal of •OH (2.7 V) is detected, which supports the enhanced $CH_3SH$ degradation performance under high humidity conditions. Notably, the signal of

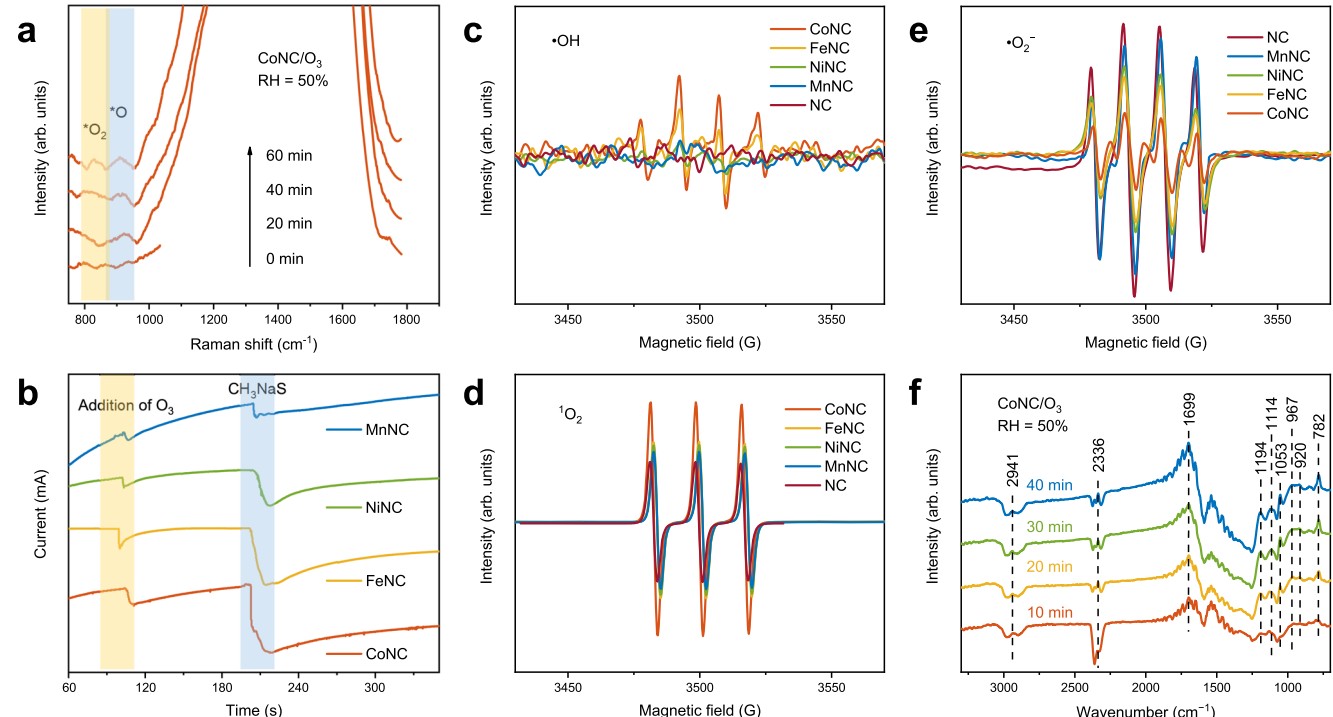

**Fig. 4 | Surface chemical reaction studies. a** In situ Raman spectra of the CoNC catalyst in the ozone ($O_3$) atmosphere. **b** The amperometric i-t curves on the MNC catalysts. **c**–**e** Electron spin resonance (ESR) spectra of 5,5-dimethylpyrroline-*N*-oxide−hydroxyl radicals (DMPO − •OH) (**c**), singlet oxygen ($^1O_2$) (**d**), and 5,5-dimethylpyrroline-*N*-oxide−superoxide radicals (DMPO − •$O_2^-$) (**e**) when the MNC catalysts exposing to $O_3$ in the dark. **f** In situ diffuse reflectance infrared Fourier transform spectroscopy (DRIFT) of the catalytic ozonation process over the CoNC.

•OH shows a metal-dependent profile. Explicitly, the CoNC exhibits the strongest •OH signal followed by the FeNC, and the NiNC, MnNC, and NC show negligible •OH signals. Similar results can be found in Fig. 4d, the $^1O_2$ concentration decreases in the order of CoNC > FeNC > NiNC > MnNC > NC. Interestingly, the generation of •$O_2^-$ exhibits the exact opposite pattern (Fig. 4e), probably owing to the competitive formation between $^1O_2$ and •$O_2^-$ (0.8 V, $E_0(^1O_2/•O_2^-)$)[49]. Notably, the order of •OH and $^1O_2$ generation concentration is consistent with the reactivity of M − *O/*$O_2$ and catalytic ozonation performance of the MNC catalysts, indicating that •OH and $^1O_2$ are regulated by the reactivity of M − *O/*$O_2$ and play a crucial role in $CH_3SH$ oxidation. In conclusion, the functional $M_1$-$N_3C_1$ active sites capture and dissociate $O_3$ molecules, forming the M − *O/*$O_2$ complexes and subsequently governing the generation of •OH/$^1O_2$/•$O_2^-$, thus achieving the efficient degradation of $CH_3SH$. Among them, the best $CH_3SH$ degradation performance of the CoNC catalyst is attributed to the most reactive Co − *O/*$O_2$ complexes and the highest concentration of •OH/$^1O_2$ with higher oxidation potential. Therefore, the reactivity of *O/*$O_2$, which has a direct or indirect impact on catalytic activity, can serve as a potential descriptor to reveal the structure-activity relationship in catalytic ozonation.

In situ diffuse reflectance infrared Fourier transform spectroscopy (DRIFT) was then performed to reveal the $CH_3SH$ oxidation pathway and the intermediates corresponding to the characteristic peaks were listed in Supplementary Table 6. When the mixture of $N_2$ and $CH_3SH$ is introduced (Supplementary Fig. 37, after $N_2$ purging), two weak bands at 2941 and 806 $cm^{-1}$ are observed, corresponding to the antisymmetric stretching mode of $CH_3$ and the stretching mode of S − O bonds, respectively[12]. This result demonstrates that the stable M − *$O_2$ complexes formed during air exposure prior to the tests exert a mild oxidizing effect on $CH_3SH$. With continuous $O_3$ flow (Fig. 4f and Supplementary Fig. 38), the emergence of bands representing the deep oxidation products, such as $SO_4^{2-}$ (1114 and 1194 $cm^{-1}$), $CO_2$ (2336 $cm^{-1}$), $SO_3^{2-}$ (967 $cm^{-1}$), and $C_2H_6O_2S$ (920 $cm^{-1}$), indicates that the reactive oxygen species generated by $O_3$ decomposition achieve the deep oxidation of $CH_3SH$[13,14]. The gradual accumulation of $SO_4^{2-}$ and $CO_2$ with $O_3$ exposure time reveals the complete mineralization of $CH_3SH$ on the $M_1$-$N_3C_1$ active sites (Supplementary Fig. 39). Moreover, the accumulation of these products is the main reason contributing to the decreased activity of the MNC catalysts[14]. Notably, during the catalytic ozonation of the NC, only a few weak characteristic peaks representing $CH_3OH$ (1051 $cm^{-1}$) and $SO_3^{2-}$ (957 $cm^{-1}$) are observed (Supplementary Fig. 40), indicating the limited performance of the NC in catalytic ozonation. This finding further emphasizes the pivotal role of the $M_1$-$N_3C_1$ as the active sites in catalytic ozonation reactions. The $CH_3SH$ oxidation pathway on the $M_1$-$N_3C_1$ active sites can be presented by the following equation (Eq. 5).

$$M - {}^*O/{}^*O_2/•OH/{}^1O_2/•O_2^- + CH_3SH \rightarrow C_2H_6O_2S \cdots CH_4O \rightarrow SO_4^{2-} + CO_2 + H_2O \tag{5}$$

## Mechanistic insights by density functional theory calculations

Density functional theory (DFT) calculations were performed to attain a reliable understanding of the structure-activity relationship on the $M_1$-$N_3C_1$ active sites. The $M_1$-$N_3C_1$ units induce the locally polarized M − C bonds to capture $O_3$ molecules onto the M atoms. Therefore, the M atoms are confirmed as the $O_3$ adsorption sites (Supplementary Figs. 41, 42 and Supplementary Table 7, M − *$O_3$, $E_{ads}$ (Mn −3.09 eV, Fe −2.27 eV, Co −1.44 eV, Ni −1.00 eV)). The limiting reaction barrier, which can be evaluated by the free energy of the rate-determining step, is an important parameter affecting the catalytic performance[31]. The Gibbs free energy for each elementary step (Eqs. 1–3) in $O_3$ decomposition is calculated to unravel the mechanism for the metal-dependent $O_3$ decomposition performance (Fig. 5a, b and Supplementary Table 8). The most endothermic steps for the $M_1$-$N_3C_1$ active

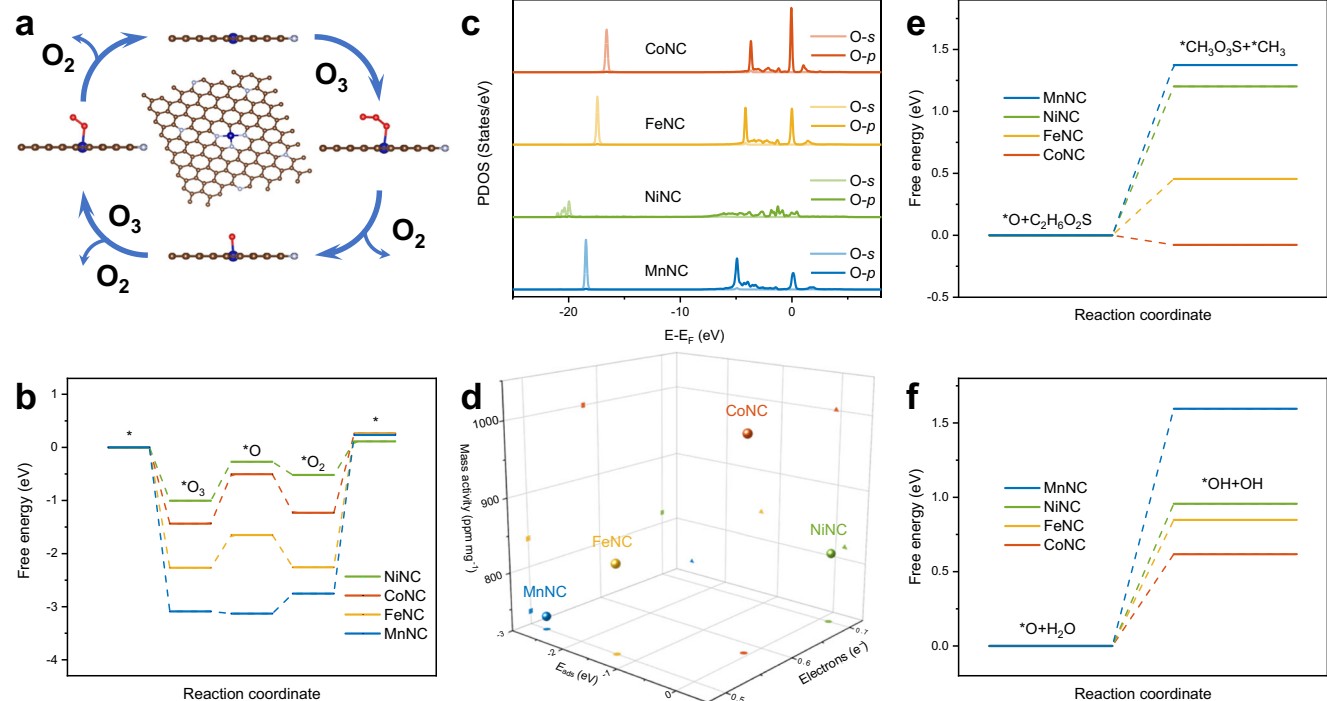

**Fig. 5 | Mechanistic insights by density functional theory calculations.**
**a** Proposed mechanism of ozone ($O_3$) decomposition over the MNC catalysts.
**b** Free energy diagrams along the MNC catalysts catalytic $O_3$ decomposition pathway. **c** Partially density of states (PDOS) of the M − *O complexes.
**d** Relationship between the adsorption energy ($E_{ads}$) and the obtained electrons of surface atomic oxygen (*O) in the M − *O complexes and the mass activities of the MNC catalysts. **e** Free energy diagrams for the reactions of *O with dimethyl sulfone ($C_2H_6O_2S$) molecules over the MNC catalysts. **f** Free energy diagrams for the reactions of *O with water ($H_2O$) molecules over the MNC catalysts. All lengths are given in Å. The blue, silver, brown, and red balls denote M, N, C, and O atoms, respectively.

sites are the dissociation of the M − *$O_2$ complexes with the limiting barrier of MnNC (2.99 eV) > FeNC (2.51 eV) > CoNC (1.49 eV) > NiNC (0.63 eV). Therefore, the different interactions of the single metal atoms and *$O_2$ lead to the differential dissociation barriers of the M − *$O_2$ complexes, thus resulting in the metal-dependent $O_3$ decomposition efficiency. The electronic property of the $Ni_1$-$N_3C_1$ active site effectively lowers the dissociation barrier of the *$O_2$, thus boosting the $O_3$ decomposition. Moreover, it is confirmed that the decrease in the $O_3$ decomposition efficiencies of the MNC catalysts over time is mainly due to the stable *$O_2$ covering the $M_1$-$N_3C_1$ active sites (Fig. 3a)[16].

For catalytic ozonation, it is predicted that the *O with high oxidation potential, rather than the *$O_2$ with mild oxidation capacity, plays a crucial role in directly influencing the catalytic ozonation performance[11]. The lowest-energy adsorption configurations (Supplementary Fig. 43) of the M − *O complexes show that the geometric and electronic states of *O are intrinsically governed by the $M_1$-$N_3C_1$ active sites. Specifically, the $M_1$-$N_3C_1$ active sites with higher electron density (Supplementary Fig. 16) shuttle more electrons to *O (Ni 0.70$e^-$ > Co 0.58$e^-$ > Fe 0.50$e^-$ > Mn 0.49$e^-$), leading to the stronger activity of *O. Therefore, the M − *O complexes with more electrons show longer M − *O bonds (Ni − *O 1.86 Å > Co − *O 1.75 Å > Fe − *O 1.67 Å > Mn − *O 1.61 Å) and higher adsorption energies ($E_{ads}$, Ni +0.15 eV > Co −0.08 eV > Fe −1.33 eV > Mn −2.68 eV), which means that the stability of the M − *O become weaker. Particularly, although the Ni − *O obtains the most electrons (0.70$e^-$), its weakest stability ($E_{ads}$ + 0.15 eV) leads to the rapid quenching of *O. These results are evidenced by the PDOS results of the M − *O. As shown in Fig. 5c, the Co − *O shows the highest density state near the Fermi level followed by the Fe − *O and Mn − *O, and the Ni − *O shows the negligible density state. Therefore, the reactivity of *O is jointly controlled by the activity and stability of *O, as can be inferred.

The mutual effect of the stability ($E_{ads}$) and activity (obtained electrons) of *O on the catalytic ozonation performance is established in Fig. 5d to evaluate the feasibility of the *O reactivity as the key descriptor for catalytic ozonation performance on the specific $M_1$-$N_3C_1$ configuration. For the MnNC, FeNC, and CoNC, there is a linear correlation between the mass activity of the catalysts and the activity and stability of *O, respectively, implying that stronger activity and weaker stability contribute to greater reactivity of *O, thus achieving better catalytic ozonation performance. In contrast, although the Ni − *O shows the highest activity (0.70$e^-$), the rapid quenching of *O is the main reason for the weak catalytic ozonation performance of the NiNC. Therefore, the reactivity of *O reveals the structure-activity relationship in catalytic ozonation on the $M_1$-$N_3C_1$ active sites.

To visually demonstrate the role of *O as a descriptor, the Gibbs free energy was calculated for both the oxidation reactions in which it directly participates and the reactions in which it mediates the formation of other reactive oxygen species. According to the results of in situ DRIFT and PTR-TOF-MS, the oxidation of dimethyl sulfone ($C_2H_6O_2S$) exhibits a slow reaction rate and is thus presumed to be the rate-limiting step for the catalytic ozonation for $CH_3SH$ degradation. As shown in Fig. 5e, the free energy for the reaction of *O with $C_2H_6O_2S$ decreases in the order of MnNC (1.37 eV) > NiNC (1.20 eV) > FeNC (0.45 eV) > CoNC (−0.08 eV), indicating that the *O can directly participate in the oxidation reactions and control the catalytic ozonation performance. The free energy for the reaction between the *O and $H_2O$ to form •OH was further calculated in Fig. 5f. The CoNC catalyst exhibits the lowest •OH formation barrier (0.62 eV), followed by FeNC (0.85 eV), NiNC (0.96 eV), and MnNC (1.60 eV), which is consistent with the ESR results and the catalytic performance. As such, the metal-dependent catalytic ozonation performance relies on the reactivity of the M − *O complexes. The highest catalytic ozonation performance with the $Co_1$-$N_3C_1$ active sites can be understood as a result that the

single Co atoms achieve the optimal binding of *O (high activity and proper stability), exhibiting the highest reactivity of *O, thus lowering the reaction barriers for the oxidation of $C_2H_6O_2S$ and maximizing the generation of reactive oxygen species (•OH and $^1O_2$) with high oxidation potentials. It is worth mentioning that the weak adsorption of S-containing byproducts such as $SO_4^{2-}$ on the $M_1$-$N_3C_1$ active sites is one of the reasons for the robust catalytic stability of the MNC catalysts (Supplementary Table 7).

## Discussion

In summary, the $O_3$ decomposition and catalytic ozonation performance were studied on a series of MNC catalysts (M: Mn, Fe, Co, and Ni) with the well-defined $M_1$-$N_3C_1$ active sites, which offers a way to unambiguously investigate the structure-activity relationships. It is found that the $M_1$-$N_3C_1$ active sites induce the locally polarized M − C bonds to capture $O_3$ molecules onto the M atoms. Subsequently, the M centers with different electronic states work as electron shuttles to determine the reactivity of the surface reaction intermediates (*O/*$O_2$), thus controlling the performance of the $O_3$ decomposition and catalytic ozonation. The best catalytic ozonation performance of the CoNC can be understood as a result that the single Co atoms achieve the optimal binding of *O (high activity and proper stability), exhibiting the highest reactivity of *O, thus lowing the reaction barriers for the oxidation of $C_2H_6O_2S$ and maximizing the generation of reactive oxygen species (•OH and $^1O_2$) with high oxidation potentials. In contrast, the lowest dissociation barrier of *$O_2$ endows the NiNC with the best $O_3$ decomposition performance. Therefore, the dissociation barrier of *$O_2$ and the reactivity of *O are proposed as potential descriptors of the $O_3$ decomposition and catalytic ozonation activity, respectively, to help design high performance catalysts.

## Methods

### Chemicals

All the chemicals were of analytical grade and used without further purification. Dicyandiamide ($C_2H_4N_4$), trimesic acid ($C_9H_6O_6$), ferrous (II) chloride ($FeCl_2$), nickel (II) chloride hexahydrate ($NiCl_2 \cdot 6H_2O$), manganese (II) chloride tetrahydrate ($MnCl_2 \cdot 4H_2O$), cobalt (II) chloride hexahydrate ($CoCl_2 \cdot 6H_2O$), commercial manganese dioxide ($MnO_2$), 2,2,6,6-tetramethyl-4-piperidone hydrochloride (TEMP), 5,5-dimethyl-pyrroline-$N$-oxide (DMPO), dimethyl sulfoxide (DMSO), sodium thiomethoxide ($CH_3NaS$), sodium sulfate ($Na_2SO_4$), cobalt phthalocyanine (CoPc), and potassium iodide (KI, spectrum pure) were purchased from Aladdin Company. Ultrapure water was used in all the experiments.

### Synthesis of the MNC single-atom catalysts

$MnCl_2 \cdot 4H_2O$ (10.6 mg), $FeCl_2$ (6.8 mg), $CoCl_2 \cdot 6H_2O$ (12.8 mg), and $NiCl_2 \cdot 6H_2O$ (12.8 mg) were respectively mixed homogeneously with a mixture of trimesic acid (500 mg) and dicyandiamide (5 g)[27]. Then, the mixture was carbonized at 800 °C for 3 h at a heating rate of 5 °C min$^{-1}$ with a high-purity $N_2$ flow. The products were correspondingly named as MnNC, FeNC, CoNC, and NiNC. The nitrogen-doped carbon material, denoted as NC, was synthesized without the addition of any metal precursors while keeping the other processing parameters constant.

### Characterization

Scanning electron microscope (SEM, Quanta 400 F, France), transmission electron microscope (TEM, FEI Tecnai G2 Spirit, Netherlands), and aberration-corrected high-angle annular dark-field scanning transmission electron microscope (AC HAADF-STEM, EM-ARM300F, Japan) were used to analyze the morphology. Brunner-Emmet-Teller (BET, JW-BK200C, China) was used to analyze the specific surface area

and pore structure. Powder X-ray diffraction (XRD) with Cu Kα radiation (Ultima IV, Rigaku Co., Japan) was used to analyze the crystal structure and phase composition. The X-ray absorption fine structure (XAFS) including X-ray absorption near-edge structure (XANES) and extended X-ray absorption fine structure (EXAFS) of CoNC at Co K-edge was collected at the Beamline of TPS44A1 in National Synchrotron Radiation Research Center (NSRRC), Taiwan. Raman spectra were attained using a Laser confocal Raman Spectrometer (inVia Qontor, Renishaw plc, England) equipped with a 633 nm laser under different atmospheres. X-ray photoelectron spectroscopy (XPS, EscaLab 250, Thermo Fisher, America) was used to analyze the elemental composition and content. Electron spin resonance (ESR, Bruker EMXplus, Bruker, Germany) was used to analyze the radicals spin-trapped by DMPO and TEMP. Inductively coupled plasma optical emission spectroscopy (ICP-OES, Agilent 5110, USA) was used to determine the metal loadings. The electrochemical tests were measured in a three-electrode quartz cell system including a saturated calomel electrode (SCE) as the reference electrode, platinum plate as the counter electrode, stainless steel coated with samples as the working electrode, and 0.5 mol L$^{-1}$ of $Na_2SO_4$ as the electrolyte. After the circuit was turned on, 1 mL of saturated $O_3$ solution and 1 mL of $CH_3NaS$ solution (0.4% in water) were added in sequence.

### Ozone decomposition and catalytic ozonation tests

Methyl mercaptan ($CH_3SH$), with a very low odor threshold, is one of the important components of VOCs emitted into the atmosphere by oil/coal/chemical industries, posing a threat to environmental safety and human health[9,43,50]. The degradation of $CH_3SH$ is of great scientific and practical importance. Therefore, $CH_3SH$ is selected as the target pollutant in our work.

$O_3$ decomposition and catalytic ozonation for $CH_3SH$ degradation tests were conducted with a continuous-flow fixed-bed reactor (the schematic diagram is shown in Supplementary Fig. 44) of the samples at room temperature (25 °C) and the inlet gas flow rate of 100 mL min$^{-1}$ [51]. 5 mg of the sample was loaded into the fixed-bed reactor and immobilized by quartz wool. The humidity of inlet gas was controlled by a humidity generator. 500 ppm of $O_3$ was generated by an $O_3$ generator (YDG, YE-TG-02PII), and the flow rate of $O_3$ in the system was controlled at 10 mL min$^{-1}$. The inlet ($C_0$) and outlet (C) concentrations of $O_3$ were continuously monitored via an $O_3$ sensor (2B Model, 106-M). The inlet concentration of $CH_3SH$ was maintained at 50 ppm by diluting 1000 ppm of $CH_3SH$ (balanced using $N_2$) with clean air. The inlet ($C_0$) and outlet (C) concentrations of $CH_3SH$ were continuously monitored via a $CH_3SH$ sensor (Detcon, DM-400IS). The outlet gas was collected with gas bags and the gas compositions were identified by Proton Transfer Reaction Time-of-Flight Mass Spectrometer (PTR-TOF-MS, PTR-TOF 1000, Ionicon Analytik GmbH, Austria).

### In situ Raman

In situ Raman spectra were attained using a Laser confocal Raman Spectrometer (inVia Qontor, Renishaw plc, England) equipped with a 633 nm laser under different atmospheres[51]. 10 mg of the sample was placed in an in situ reaction cell. The whole system was first purged with $N_2$ for 20 min before each experiment. The inlet $O_3$ was generated using an $O_3$ generator (CH-ZTW3G, Chuanghuan Ozocenter) with a concentration of 100 ppm. The flow rate was set as 40 mL min$^{-1}$ during the spectra acquisition process. The humidity level was set to 50%.

### In situ diffuse reflectance infrared Fourier transform spectroscopy

In situ diffuse reflectance infrared Fourier transform spectroscopy (DRIFTS) was conducted in the range of 600 − 4000 cm$^{-1}$ using a TENSOR II Fourier transform infrared (FTIR) spectrometer (EQUINOX

55, Bruker, Germany)[51]. 10 mg of the sample was mixed homogeneously with 1 g of KI and the mixture was irradiated under an infrared lamp for 20 min to remove the adsorbed water. Subsequently, the mixture was placed in a Harrick Scientific Praying Mantis DRIFTS cell and the surface was leveled. Before each experiment, the whole system was purged with $N_2$ for 30 min. The background spectrum was collected in a gas stream containing 50 mL min$^{-1}$ of $N_2$. Next, the catalyst underwent the adsorption process with 50 ppm of $CH_3SH$ under an $N_2$ atmosphere for 40 min and then underwent the catalytic ozonation process under an $O_3$ atmosphere (50 ppm) for 40 min at 50% humidity.

### Density functional theory calculations

All the spin-polarized DFT-D2 calculations were conducted in the "Vienna ab initio simulation package" (VASP 5.4) with the generalized gradient approximation[52–54]. Valence electron density was expanded in a plane wave basis set with a 400 eV cutoff for the kinetic energy and the projector augmented wave method was used to describe the interactions between core and valence electrons. A 127-atom supercell slab (C 119, N 7, M 1) with lattice parameters of $19.7 \times 17.0 \times 20.0$ Å$^3$ was used to model the samples. In addition, 0.2 eV was taken as the Gaussian smearing width. And $3 \times 3 \times 1$ K points were set in the Brillouin zone. In all calculations, the positions of the atoms were allowed to relax until all forces were smaller than 0.01 eV Å$^{-1}$. The adsorption energy ($E_{ads}$) and the Gibbs free energy ($G$) were calculated by the following formula:

$$E_{ads} = E_t - (E_z + E_m) \tag{6}$$

$$G = E_t + E_{ZPE} + nRT - TS \tag{7}$$

where $E_t$, $E_z$, $E_m$, and $E_{ZPE}$ represent the energy of the adsorption complex, catalyst, individual molecule, and zero-point energy, respectively.

## Data availability

The experimental data supporting the findings of this study are available within the article, Supplementary Information, and Source Data. Additional data are available from the corresponding authors upon request. Source data are provided with this paper.

## Code availability

Only the commercial codes were used in this work (See references).

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

## Acknowledgements

We acknowledge financial support from the National Natural Science Foundation of China (41603097 (D.X.), 21673086 (D.X.), 51872341 (S.L.)), Guangdong Basic and Applied Basic Research Foundation (2022B1515020097 (D.X.)), Opening Fund of the State Key Laboratory of Environmental Geochemistry (SKLEG2022221 (D.X.)), and Fundamental Research Funds for the Central Universities, Sun Yat-sen University (22lgqb21 (D.X.)).

## Author contributions

D.X., S.L., and J.Y. supervised the project. D.M. and D.X. initiated the research. D.M., Y.Z., Y.H., and X.G. synthesized and characterized the materials. D.M. performed the theoretical calculations. D.M., Qiy. L., Qiw. L., C.H., and D.X. analyzed the data. D.M. wrote the paper with inputs from all authors.

## Competing interests

The authors declare no competing interests.
