## [Peer Review File · Nature Communications]

Catalytic ozonation mechanism over M1-N3C1 active sitesREVIEWER COMMENTS

Reviewer #1 (Remarks to the Author):

This manuscript reported a series of MNC catalysts with well-defined M₁-N₃C₁ active sites for catalytic ozonation. The surface chemical states, electronic structure, and catalytic activity of the MNC catalysts were comprehensively studied. More importantly, this manuscript was the first to identify the reactivity of surface atomic oxygen as a descriptor for the structure-activity relationship in the catalytic ozonation. Overall, this manuscript is well-written and holds significant inspiration for the field of the catalytic ozonation and heterogeneous catalysis. After a review, this manuscript can be accepted after minor revision, by answering the following questions:

1. As stated, the central metal atoms can affect the polarized charge density distribution of the M-C bond among the various M₁-N₃C₁, so what is the possible reason for that?
2. A strong signal of •OH is observed, especially in the CoNC system, so how the •OH was produced? Was H₂O involved in the reaction?
3. The strength patterns of •OH and [•]O is consistent with degradation performance among the various M₁-N₃C₁, so does it mean that the •OH and [•]O are the primary contributors to the degradation of CH₃SH?
4. In page 12, line 238-243, the CoNC has a much better mass-activity of 1000 ppm mg⁻¹ over the others, what are the possible reasons?
5. An appropriate explanation is suggested to provide for explaining the decrease of the stability of CoNC after 1000 mins running.
6. Figures S11 and S12. The boundaries of all built model boxes in these diagrams should be drawn to show their periodic structure.
7. Please carefully examine and correct the grammatical and spelling errors.

Reviewer #2 (Remarks to the Author):

In this work, the O₃ decomposition and catalytic ozonation performance were studied on a series of MNC catalysts (M: Mn, Fe, Co, and Ni) with the well-defined M₁-N₃C₁ active sites. The combined in situ characterization and theoretical calculations reveal the metal-dependent catalytic activity, with the reactivity of surface atomic oxygen (*O) being first identified as a descriptor for the structure-activity relationship in the catalytic ozonation. Additionally, the dissociation barrier of surface peroxide species (*O₂) is proposed as a descriptor for the structure-activity relationship in the O₃ decomposition. Overall, the results are interesting and can provide a reference to the design of high-performance catalytic ozonation catalysts. Therefore, I would like to recommend its publication in Nature Communications after revisions. The specific concerns are listed as follows:

1. Does ambient humidity affect the ozone removal performance of MNC catalysts? The current manuscript only tested the ozone removal performance of the catalysts at 50% RH. In addition, ozone

removal tests under dry and higher humidity conditions should be supplemented.

2. For the characterization tests (e.g., in situ Raman and DRIFT) and DFT calculations, did the authors consider the influence of water molecules? This is puzzling to me because the tests in Fig.3 were all conducted under humid conditions.

3. Line 186. It was mentioned that the catalysts exhibit the decreasing O₃ decomposition efficiencies after running for 30 min, which may be owing to the occupation of active sites by intermediates. So, is there a way to restore the activity of catalysts?

4. The increase of CH₃SH degradation efficiencies of the catalyst at >90 % RH is an interesting phenomenon, so a more detailed explanation is recommended.

5. Related tests should be supplemented to verify the stability of the catalysts after the catalytic reaction.

Reviewer #3 (Remarks to the Author):

In this manuscript, Ma et al. presented some results in synthesis of M₁-N₃C-based materials for catalytic ozonation of sulfur-contained compounds and the mechanism. In the work, the authors tried to analyze the active sites for the reaction of O₃ activation and organic compound decomposition. However, the manuscript suffers from fatal deficiencies and the mechanism was not well illustrated and conclusive. The manuscript was not suggested to be accepted by Nat Commun. Other comments are listed below.

1. It is well known that carbon is an active catalyst for ozonation and functional groups on carbon also affect the reaction. However, no investigation was conducted. Thus, it is not clearly to derive the major role of M-N-C in catalysis.

2. The authors should test the control carbon sample in catalysis.

3. The authors did not suggest the detailed metal chemical state in the samples by XPS.

4. It is also well known that carbon-based catalytic ozonation will produce different reactive species and the authors did not investigate their reactions with sulfur compound. Thus, it is difficult to link the active site to organic decomposition.

5. In N₂ atmosphere, it was suggested that mild oxidation of CH₃SH by the M-O₂* complexes. How will the complexes be generated?

6. In DFT calculations, Eads of different metals suggested the strongest adsorption of O₃ on Mn, which indicates the favored thermodynamics. However, Mn-based material was not good in catalysis. Other calculations seem not to agree with the tests of different metals.

7. The conclusion on O* and *O₂ as descriptors for catalytic ozonation and O₃ decomposition was not convincing as the authors did not well consider other species involved in catalytic ozonation.

Manuscript ID: NCOMMS-23-17636

Title: Catalytic ozonation mechanism over $M_1-N_3C_1$ active sites

Authors: Dingren Ma, Qiyu Lian, Yexing Zhang, Yajing Huang, Xinyi Guan, Qiwen Liang, Chun He, Dehua Xia, Shengwei Liu, and Jiaguo Yu

General note: The comments from reviewers are presented in black *italic* font style. Responses to each comment are formatted with indentation and displayed in regular style. The “revised text as it appears in the manuscript” is presented in blue and indented. The reference numbers provided in this document are specific to the references cited herein and may not correspond with those used in the manuscript or supplementary information. This document contains the responses to the comments from the three reviewers.

Reviewer #1

General Comment: *This manuscript reported a series of MNC catalysts with well-defined $M_1-N_3C_1$ active sites for catalytic ozonation. The surface chemical states, electronic structure, and catalytic activity of the MNC catalysts were comprehensively studied. More importantly, this manuscript was the first to identify the reactivity of surface atomic oxygen as a descriptor for the structure-activity relationship in catalytic ozonation. Overall, this manuscript is well-written and holds significant inspiration for the field of catalytic ozonation and heterogeneous catalysis. After a review, this manuscript can be accepted after minor revision, by answering the following questions:*

Reply: We express our gratitude to the reviewer for acknowledging the research value and providing constructive comments to enhance the quality of this work. We have answered the reviewer's comments point by point and rectified any relevant issues. We would greatly appreciate it if the reviewer could reassess our revised manuscript for potential publication in the journal.

Comment 1: *As stated, the central metal atoms can affect the polarized charge density distribution of the M-C bond among the various $M_1-N_3C_1$, so what is the possible reason for that?*

Reply: We express our gratitude to the reviewer for this comment. The variation in the polarized charge density distribution of the M–C bond can be attributed to the different energy states of the electrons in the 3d orbitals of the various single metal atoms and the orbital interactions between the single metal atoms and C/N atoms.

It has been reported that the electronic structure of active sites in single-atom catalysts is determined by both the single metal atoms and their coordination environments.¹ Therefore, the electronic structure of the $M_1-N_3C_1$ active sites is solely governed by single metal atoms (M: Mn, Fe, Co, and Ni) within the well-defined N_3C_1 coordination in our work. Considering the different energy states of

the electrons in the 3d orbitals of the various single metal atoms and the orbital interactions between the single metal atoms and C/N atoms, the polarized charge density distribution of the M–C bonds exhibits a dependence on the nature of the metal atom.² This is confirmed by the density functional theory (DFT) calculations (Supplementary Fig. 15–16).

To clarify this result, we have revised the manuscript description as follows.

Line 92, “The M centers, with different 3d orbital electrons acting as the electron shuttles, regulate the structural and electronic states of *O/*O₂ as well as the mechanisms and kinetics for both the nonradical (*O, *O₂, and ¹O₂) and radical (•O₂⁻ and •OH) pathways, thus controlling the catalytic ozonation performance.^{1, 3}”

Line 176, “The calculated projected density of states (PDOS) results in Supplementary Fig. 15 show that the metal 3d orbitals exhibit significant orbital electronic coupling to the C 2p and N 2p orbitals, respectively, confirming the stability of the M₁-N₃C₁ coordination.⁴”

Line 190, “More importantly, different electronic states are observed among metal centers sharing the same N₃C₁ coordination environment (Supplementary Fig. 16), which can be attributed to the different energy states of the 3d orbital electrons and orbital interactions between single metal atoms and C/N atoms (Supplementary Fig. 15).”

Supplementary Figure 15. Partially density of states (PDOS) of the MNC catalysts (a–d MnNC, FeNC, CoNC, and NiNC).

Supplementary Figure 16. Charge difference isosurfaces and Bader charge of the $M_1-N_3C_1$ units (**a–d** $Mn_1-N_3C_1$, $Fe_1-N_3C_1$, $Co_1-N_3C_1$, and $Ni_1-N_3C_1$). Isosurfaces level = 0.005. All lengths are given in Å. The purple, yellow, blue, silver gray, brown, and silver balls denote Mn, Fe, Co, Ni, C, and N atoms, respectively. The blue and yellow isosurfaces represent charge accumulation and depletion in the space, respectively.

References:

1. Zhang, J. et al. Tuning the coordination environment in single-atom catalysts to achieve highly efficient oxygen reduction reactions. *J. Am. Chem. Soc.* **141**, 20118-20126 (2019).
2. Chang, Q. et al. Metal-coordinated phthalocyanines as platform molecules for understanding isolated metal sites in the electrochemical reduction of CO_2 . *J. Am. Chem. Soc.* **144**, 16131-16138 (2022).
3. Huang, Y. et al. Enhanced catalytic ozonation for eliminating CH_3SH via graphene-supported positively charged atomic Pt undergoing Pt^{2+}/Pt^{4+} redox cycle. *Environ. Sci. Technol.* **55**, 16723-16734 (2021).
4. Wang, C. et al. Co and Pt dual-single-atoms with oxygen-coordinated Co-O-Pt dimer sites for ultrahigh photocatalytic hydrogen evolution efficiency. *Adv. Mater.* **33**, e2003327 (2021).

Comment 2: A strong signal of $\bullet OH$ is observed, especially in the CoNC system, so how the $\bullet OH$ was produced? Was H_2O involved in the reaction?

Reply: We express our gratitude to the reviewer for this comment. In our study, the $\bullet OH$ is generated through the reaction between surface atomic oxygen ($*O$) and

H₂O molecules. The presence of H₂O molecules is a prerequisite for the generation of •OH.

It has been reported that the active intermediate oxygen species generated during O₃ decomposition can react with H₂O molecules to generate •OH.¹ Song et al. conducted a more comprehensive investigation into the generation of •OH, which was attributed to the protonation reaction between *O and H₂O molecules.² Our density functional theory (DFT) calculations offer additional support for the occurrence of this reaction. As shown in Fig. 5f, the CoNC catalyst exhibits the lowest •OH formation barrier (0.62 eV), followed by FeNC (0.85 eV), NiNC (0.96 eV), and MnNC (1.60 eV), which is consistent with the electron spin resonance (ESR) results (Fig. 4c). The ESR results show that the signal of •OH can be detected when using dimethylpyridine N-oxide (DMPO) as a trapping agent, and this signal exhibits a metal-dependent profile. Explicitly, the CoNC exhibits the strongest •OH signal followed by the FeNC, while the NiNC, MnNC, and NC show negligible signals. Therefore, the •OH is generated through the reaction between *O and H₂O molecules.

To clarify this result, we have revised the manuscript description as follows.

Line 64, “Recent studies have proposed that in catalytic ozonation reactions, the *O and *O₂ not only directly participate in oxidation reactions but also regulate the generation of other reactive oxygen species, such as hydroxyl radicals (•OH), singlet oxygen (¹O₂), and superoxide radicals (•O₂⁻), by reacting with H₂O, thus indirectly affecting catalytic activity.^{3,4}”

Line 325, “Besides being directly involved in oxidation reactions, the *O/*O₂ also regulates the generation of other reactive oxygen species (•OH, ¹O₂, and •O₂⁻) by reacting with H₂O.²”

Line 430, “The free energy for the reaction between the *O and H₂O to form •OH was further calculated in Fig. 5f. The CoNC catalyst exhibits the lowest •OH formation barrier (0.62 eV), followed by FeNC (0.85 eV), NiNC (0.96 eV), and MnNC (1.60 eV), which is consistent with the ESR results and the catalytic performance.”

Figure 4. **Surface chemical reaction studies.** Electron spin resonance (ESR) spectra of DMPO-•OH (c) when the MNC catalysts exposing to O₃ in the dark.

Figure 5. **Mechanistic insights by DFT calculations.** f Free energy diagrams for the reactions of *O with H₂O molecules over the MNC catalysts.

References:

1. Sun, Z. B., Si, Y. N., Zhao, S. N., Wang, Q. Y. & Zang, S. Q. Ozone decomposition by a manganese-organic framework over the entire humidity range. *J. Am. Chem. Soc.* **143**, 5150-5157 (2021).
2. Wang, Y. et al. Occurrence of both hydroxyl radical and surface oxidation pathways in N-doped layered nanocarbons for aqueous catalytic ozonation. *Appl. Catal. B* **254**, 283-291 (2019).
3. Ren, T., Yin, M., Chen, S., Ouyang, C., Huang, X. & Zhang, X. Single-atom Fe-N₄ sites for catalytic ozonation to selectively induce a nonradical pathway toward wastewater purification. *Environ. Sci. Technol.* **57**, 3623-3633 (2023).
4. Yu, G. et al. Insights into the mechanism of ozone activation and singlet oxygen generation on N-doped defective nanocarbons: a DFT and machine learning study. *Environ. Sci. Technol.* **56**, 7853-7863 (2022).

Comment 3: *The strength patterns of •OH and •O₂⁻ are consistent with degradation performance among the various MNC, so does it mean that the •OH and •O₂⁻ are the primary contributors to the degradation of CH₃SH?*

Reply: We express our gratitude to the reviewer for this comment. Notably, the degradation performance of the various MNC is consistent with the strength patterns of •OH and ¹O₂, rather than •O₂⁻. The •OH and ¹O₂, with the high oxidation potentials, can participate in the oxidation reaction of CH₃SH.¹ However, this does not imply that the •OH and ¹O₂ are the primary contributors to CH₃SH degradation. The oxidation reaction also involves surface peroxide species (*O₂, 1.35 V) and surface atomic oxygen (*O, 2.43 V).²

Recent studies have proposed that in catalytic ozonation reactions, the *O and *O₂ not only directly participate in oxidation reactions but also regulate the generation of other reactive oxygen species, such as hydroxyl radicals (•OH), singlet oxygen (¹O₂), and superoxide radicals (•O₂⁻), by reacting with H₂O, thus indirectly affecting

catalytic activity.³ We first studied the reactivity of $^*O/^*O_2$ by amperometric i-t curve (i-t) tests. As shown in Fig. 4b, the sequential addition of saturated O_3 solution and sodium thiomethoxide (CH_3NaS) solution shows a significant current impulse, indicating the formation of $M-^*O$ and/or $M-^*O_2$ complexes and the subsequent oxidation of CH_3NaS .⁴ The reactivity of the complexes ($M-^*O/^*O_2$, based on the degree of the current impulse in Supplementary Fig. 36) relies on the variety of single metal atoms following the order of $CoNC > FeNC > NiNC > MnNC$, which is consistent with the catalytic ozonation performance (Fig. 3b). This result confirms the direct involvement of $M-^*O/^*O_2$ in the catalytic ozonation reactions and highlights the direct effects of $M-^*O/^*O_2$ reactivity on catalytic ozonation performance. Moreover, the presence of other reactive oxygen species was studied by the electron spin resonance (ESR) tests (Fig. 4c-e). It is found that the concentration order of the generated $\bullet OH$ and 1O_2 is consistent with the reactivity of $M-^*O/^*O_2$ and catalytic ozonation performance of the MNC catalysts, indicating that the $\bullet OH$ and 1O_2 are regulated by the reactivity of $M-^*O/^*O_2$ and play a crucial role in the CH_3SH oxidation. Therefore, we propose that the functional $M_1-N_3C_1$ active sites capture and dissociate O_3 molecules, forming the $M-^*O/^*O_2$ complexes and subsequently governing the generation of $\bullet OH/^1O_2/\bullet O_2^-$, thus achieving the efficient degradation of CH_3SH . Furthermore, we suggest that the reactivity of $^*O/^*O_2$, which has a direct or indirect impact on catalytic activity, can serve as a potential descriptor to reveal the structure-activity relationship in catalytic ozonation. The density functional theory (DFT) calculations further confirmed these findings. The Gibbs free energy was calculated for both the oxidation reactions in which the *O directly participates and the reactions in which the *O mediates the formation of other reactive oxygen species. The results (Fig. 5e-f) confirm that the *O not only directly participates in oxidation reactions but also regulates the generation of other reactive oxygen species, thus governing catalytic activity.

To clarify this result, we have revised the manuscript description as follows.

Line 64, “Recent studies have proposed that in catalytic ozonation reactions, the *O and *O_2 not only directly participate in oxidation reactions but also regulate the generation of other reactive oxygen species, such as hydroxyl radicals ($\bullet OH$), singlet oxygen (1O_2), and superoxide radicals ($\bullet O_2^-$), by reacting with H_2O , thus indirectly affecting catalytic activity.^{5,6}”

Line 315, “The reactivity of $^*O/^*O_2$, directly determined by the $M_1-N_3C_1$ active sites, was investigated ... can serve as a potential descriptor to reveal the structure-activity relationship in catalytic ozonation.”

Line 421, “To visually demonstrate the role of *O as a descriptor, the Gibbs free energy was ... thus lowering the reaction barriers for the oxidation of $C_2H_6O_2S$ and maximizing the generation of reactive oxygen species ($\bullet OH$ and 1O_2) with high oxidation potentials.”

Figure 3. O_3 decomposition and catalytic ozonation performance. **b** Catalytic ozonation for CH_3SH degradation tests over the NC and MNC catalysts.

Figure 4. **Surface chemical reaction studies.** **a** In situ Raman spectra of the CoNC catalyst in the O_3 atmosphere. **b** The amperometric *i-t* curves on the MNC catalysts. Electron spin resonance (ESR) spectra of $DMPO\cdot OH$ (**c**), 1O_2 (**d**), and $DMPO\cdot O_2\cdot$ (**e**) when the MNC catalysts exposing to O_3 in the dark. **f** In situ diffuse reflectance infrared Fourier transform spectroscopy (DRIFT) of the catalytic ozonation process over the CoNC.

Figure 5. **Mechanistic insights by DFT calculations.** e Free energy diagrams for the reactions of $*O$ with $C_2H_6O_2S$ molecules over the MNC catalysts. f Free energy diagrams for the reactions of $*O$ with H_2O molecules over the MNC catalysts.

Supplementary Figure 36. The corresponding current variation of the chronoamperometry curves on the MNC catalysts.

References:

1. Qu, W. et al. Self-accelerating interfacial catalytic elimination of gaseous sulfur-containing volatile organic compounds as microbubbles in a facet-engineered three-dimensional BiOCl sponge Fenton-like process. *Environ. Sci. Technol.* **56**, 11657-11669 (2022).
2. Bing, J., Hu, C. & Zhang, L. Enhanced mineralization of pharmaceuticals by surface oxidation over mesoporous γ -Ti- Al_2O_3 suspension with ozone. *Appl. Catal. B* **202**, 118-126 (2017).
3. Wang, Y. et al. Occurrence of both hydroxyl radical and surface oxidation pathways in N-doped layered nanocarbons for aqueous catalytic ozonation. *Appl. Catal. B* **254**, 283-291 (2019).
4. Wang, Y., Duan, X., Xie, Y., Sun, H. & Wang, S. Nanocarbon-based catalytic ozonation for aqueous oxidation: engineering defects for active sites and tunable reaction pathways. *ACS Catal.* **10**, 13383-13414 (2020).
5. Ren, T., Yin, M., Chen, S., Ouyang, C., Huang, X. & Zhang, X. Single-atom Fe- N_4 sites for catalytic ozonation to selectively induce a nonradical pathway toward wastewater purification. *Environ. Sci. Technol.* **57**, 3623-3633 (2023).
6. Yu, G. et al. Insights into the mechanism of ozone activation and singlet oxygen generation on N-doped defective nanocarbons: a DFT and machine learning study. *Environ. Sci. Technol.* **56**, 7853-7863 (2022).

Comment 4: *In page 12, line 238-243, the CoNC has a much better mass-activity of 1000 ppm mg⁻¹ over the others, what are the possible reasons?*

Reply: We express our gratitude to the reviewer for this comment. The superior catalytic ozonation performance of the CoNC can be attributed to the optimal binding of surface atomic oxygen (*O) by the single Co atoms, which exhibits high activity and proper stability. This results in the highest reactivity of *O, thus lowering reaction barriers for C₂H₆O₂S oxidation and maximizing the generation of reactive oxygen species (•OH and ¹O₂) with high oxidation potentials. The results have been fully confirmed across three sections: Ozone decomposition and catalytic ozonation performance, Identification of surface reaction intermediates, and Mechanistic insights by DFT calculations.

The optimal catalytic ozonation performance of the CoNC is comprehensively described in Line 435.

Line 435, “The highest catalytic ozonation performance with the Co₁-N₃C₁ active sites can be understood as a result that the single Co atoms achieve the optimal binding of *O (high activity and proper stability), exhibiting the highest reactivity of *O, thus lowering the reaction barriers for the oxidation of C₂H₆O₂S and maximizing the generation of reactive oxygen species (•OH and ¹O₂) with high oxidation potentials.”

Comment 5: *An appropriate explanation is suggested to provide for explaining the decrease in the stability of CoNC after 1000 min running.*

Reply: We express our gratitude to the reviewer for this comment. The decrease in stability observed after 1000 min of the CoNC is attributed to the deposition of by-products onto the surface of the CoNC.

We conducted characterizations of the CoNC after the reaction and found that its morphological structure remains unchanged (Supplementary Fig. 32–33). Therefore, the observed decrease in stability of the CoNC (Supplementary Fig. 31) can be attributed to the deposition of by-products onto its surface.¹ The X-ray photoelectron spectroscopy (XPS) results (Supplementary Fig. 25) indicate that a significant number of sulfur-containing by-products are deposited onto the surface of the CoNC. The finding is further confirmed by the results obtained from in situ diffuse reflectance infrared Fourier transform spectroscopy (DRIFT, Supplementary Fig. 38). Notably, the stability of the MNC catalysts is better than that of previously reported catalysts.² The exceptional catalytic stability can be attributed in part to the weak adsorption of sulfur-containing by-products on the M₁-N₃C₁ active sites (Supplementary Table 7).

To clarify this result, we have revised the manuscript description as follows.

Line 277, “Regarding the stability of the catalysts, the CoNC catalyst still retains the 97.1% CH₃SH degradation efficiency after the continuous running of 1000 min in Supplementary Fig. 31, which is better than the well-reported catalysts.¹⁻³ Moreover, the morphology and single Co atoms of the CoNC catalyst (Supplementary Fig. 32–33) remain unchanged after the reaction, confirming the exceptional stability of the CoNC catalyst.”

Line 360, “The gradual accumulation of SO₄²⁻ and CO₂ with O₃ exposure time reveals the complete mineralization of CH₃SH on the M₁-N₃C₁ active sites (Supplementary Fig. 39). Moreover, the accumulation of these products is the main reason contributing to the decreased activity of the MNC catalysts.¹”

Line 439, “It is worth mentioning that the weak adsorption of S-containing byproducts such as SO₄²⁻ on the M₁-N₃C₁ active sites is one of the reasons for the robust catalytic stability of the MNC catalysts (Supplementary Table 7).”

Supplementary Figure 25. X-ray photoelectron spectroscopy (XPS) of the CoNC in different states (a survey spectrum of the CoNC after the CH₃SH dynamic degradation test in the air for 60 min. b survey spectrum of the CoNC after the catalytic ozonation for CH₃SH degradation test for 60 min. c the S 2p spectrum of the CoNC after the CH₃SH dynamic degradation test in the air for 60 min. d the S 2p spectrum of the CoNC after the catalytic ozonation for CH₃SH degradation test for 60 min.).

Supplementary Figure 31. Long-term test of catalytic ozonation for CH₃SH degradation over the CoNC.

Supplementary Figure 32. Scanning electron microscope (**a** SEM) and aberration-corrected high-angle annular dark-field scanning transmission electron microscope (**b** AC HAADF-STEM) images of the used CoNC catalyst.

Supplementary Figure 33. **a** Powder X-ray diffraction (XRD) pattern of the used CoNC catalyst. **b** Raman spectrum of the used CoNC catalyst.

Supplementary Figure 38. In situ diffuse reflectance infrared Fourier transform (DRIFT) spectroscopy of the catalytic ozonation processes over the MnNC (a), FeNC (b), CoNC (c), and NiNC (d).

Supplementary Figure 39. The concentrations of CO_2 and SO_4^{2-} of the catalytic ozonation process over the CoNC (Figure 4f).

Supplementary Table 7. Optimize the adsorption energy corresponding to the structure of the resting point in the molecular adsorption process of the MNC catalysts.

Catalysts	Adsorption energy (eV)					
	O_3	CH_3SH	O_2	H_2O	CO_2	SO_4^{2-}
CoNC	-1.44	-0.45	-1.49	-0.32	-0.21	-0.13
FeNC	-2.27	-1.22	-2.51	-0.37	-0.20	-0.15
NiNC	-1.00	-0.35	-0.63	-0.24	-0.18	-0.15
MnNC	-3.09	-1.39	-2.99	-0.62	-0.59	-0.87
NC	-0.88	/	/	/	/	/

References:

1. Huang, Y. et al. Enhanced catalytic ozonation for eliminating CH₃SH via graphene-supported positively charged atomic Pt undergoing Pt²⁺/Pt⁴⁺ redox cycle. *Environ. Sci. Technol.* **55**, 16723-16734 (2021).
2. Ma, D., Liu, W., Huang, Y., Xia, D., Lian, Q. & He, C. Enhanced catalytic ozonation for eliminating CH₃SH via stable and circular electronic metal-support interactions of Si-O-Mn bonds with low Mn loading. *Environ. Sci. Technol.* **56**, 3678-3688 (2022).
3. Xia, D. et al. Enhanced performance and conversion pathway for catalytic ozonation of methyl mercaptan on single-atom Ag deposited three-dimensional ordered mesoporous MnO₂. *Environ. Sci. Technol.* **52**, 13399-13409 (2018).

Comment 6: *Figures S11 and S12. The boundaries of all built model boxes in these diagrams should be drawn to show their periodic structure.*

Reply: We express our gratitude to the reviewer for this comment. Following the suggestion of the reviewer, we have drawn the boundaries of all the constructed model boxes in Supplementary Fig. 13–14.

Supplementary Figure 13. Theoretical models of the MNC catalysts (a–d MnNC, FeNC, CoNC, and NiNC). Inset: top view. All lengths are given in Å. The purple, yellow, blue, silver gray, brown, and silver balls denote Mn, Fe, Co, Ni, C, and N atoms, respectively.

Supplementary Figure 14. Theoretical models of the M_1-N_4 units (**a–d** Mn_1-N_4 , Fe_1-N_4 , Co_1-N_4 , and Ni_1-N_4). Inset: top view. All lengths are given in Å. The purple, yellow, blue, silver gray, brown, and silver balls denote Mn, Fe, Co, Ni, C, and N atoms, respectively.

Comment 7: *Please carefully examine and correct the grammatical and spelling errors.*

Reply: We express our gratitude to the reviewer for this comment. Following the suggestion of the reviewer, we have carefully examined the entire manuscript and corrected any grammatical and spelling errors.

Line 47, “The simultaneous removal of SVOCs and O_3 from the air is thus of high importance.”

Line 70, “Nevertheless, an in-depth understanding of the surface chemical reactions is limited by the unclear active sites.”

Line 238, “Interestingly, the degradation efficiencies of CH_3SH by the MNC catalysts in the air (Supplementary Fig. 22) show a metal-dependent profile similar to that observed under the O_3 conditions. The order of CH_3SH degradation efficiencies achieved is $CoNC$ (84.5%) > $FeNC$ (71.0%) > $NiNC$ (41.5%) > $MnNC$ (37.7%) > NC (35.8%).”

Line 421, “To visually demonstrate the role of $*O$ as a descriptor, the Gibbs free energy was calculated for both the oxidation reactions in which it directly participates and the reactions in which it mediates the formation of other reactive oxygen species.”

Reviewer 2

General Comments: *In this work, the O₃ decomposition and catalytic ozonation performance were studied on a series of MNC catalysts (M: Mn, Fe, Co, and Ni) with the well-defined M₁-N₃C₁ active sites. The combined in situ characterization and theoretical calculations reveal the metal-dependent catalytic activity, with the reactivity of surface atomic oxygen (*O) being first identified as a descriptor for the structure-activity relationship in the catalytic ozonation. Additionally, the dissociation barrier of surface peroxide species (*O₂) is proposed as a descriptor for the structure-activity relationship in the O₃ decomposition. Overall, the results are interesting and can provide a reference to the design of high-performance catalytic ozonation catalysts. Therefore, I would like to recommend its publication in Nature Communications after revisions. The specific concerns are listed as follows:*

Reply: We express our gratitude to the reviewer for acknowledging the research value and providing constructive comments to enhance the quality of this work. We have answered the reviewer's comments point by point and rectified any relevant issues. We would greatly appreciate it if the reviewer could reassess our revised manuscript for potential publication in the journal.

Comment 1: *Does ambient humidity affect the ozone removal performance of MNC catalysts? The current manuscript only tested the ozone removal performance of the catalysts at 50% RH. In addition, ozone removal tests under dry and higher humidity conditions should be supplemented.*

Reply: We express our gratitude to the reviewer for this comment. As noted by the reviewer, the ozone (O₃) removal performance of the MNC catalyst is influenced by ambient humidity.¹ Following the reviewer's suggestion, we conducted O₃ removal tests under dry and high humidity conditions.

As shown in Fig. 3a and Supplementary Fig. 27, the O₃ removal performance of the MNC catalyst is significantly influenced by ambient humidity. Under dry conditions, the MNC catalysts exhibit gradual deactivation after 20 min, despite achieving the complete removal of O₃. The stability of the MNC catalysts gradually increases with increasing humidity levels. When the humidity exceeds 90%, the NiNC and CoNC do not experience deactivation, while the FeNC and MnNC exhibit deactivation after 50 min and 35 min, respectively. These findings are associated with the acceleration of the O₃ decomposition reaction by H₂O molecules.¹ The free energy diagram of the reaction between surface atomic oxygen (*O) and H₂O molecules on the MNC catalyst (Fig. 5f) provides the important evidence for the conclusions.

To clarify this result, we have revised the manuscript description as follows.

Line 257, “These results are consistent with the O₃ decomposition performance of the MNC catalysts under both dry and wet conditions. As shown in Supplementary Fig. 27a, under dry conditions, the MNC catalysts exhibit gradual deactivation after 20 min, despite achieving the complete removal of O₃. The stability of the MNC catalysts exhibits a gradual increase with an elevation in humidity levels (Fig. 3a and Supplementary Fig. 27b). When the humidity exceeds 90% (Supplementary Fig. 27b), the NiNC and CoNC remain active, while the FeNC and MnNC exhibit deactivation after 50 min and 35 min, respectively. These findings suggest that the H₂O molecule not only promotes O₃ decomposition, but also enhances the catalytic ozonation performance of the MNC catalysts by facilitating O₃ decomposition. Notably, the enhanced decomposition of O₃ by H₂O can be attributed to the reaction between H₂O and the intermediates resulting from O₃ decomposition. This will be discussed later in DFT calculations.”

Line 430, “The free energy for the reaction between the *O and H₂O to form •OH was further calculated in Fig. 5f. The CoNC catalyst exhibits the lowest •OH formation barrier (0.62 eV), followed by FeNC (0.85 eV), NiNC (0.96 eV), and MnNC (1.60 eV), which is consistent with the ESR results and the catalytic performance.”

Figure 3. O₃ decomposition and catalytic ozonation performance. a O₃ dynamic decomposition tests over the NC and MNC catalysts.

Figure 5. Mechanistic insights by DFT calculations. f Free energy diagrams for the reactions of *O with H₂O molecules over the MNC catalysts.

Supplementary Figure 27. Catalytic O₃ decomposition tests over the MNC catalysts under different relative humidity (RH) conditions (**a** RH < 1%. **b** RH > 90%).

References:

1. Dong, C., Yang, J. J., Xie, L. H., Cui, G., Fang, W. H. & Li, J. R. Catalytic ozone decomposition and adsorptive VOCs removal in bimetallic metal-organic frameworks. *Nat. Commun.* **13**, 4991 (2022).

Comment 2: For the characterization tests (e.g., in situ Raman and DRIFT) and DFT calculations, did the authors consider the influence of water molecules? This is puzzling to me because the tests in Fig.3 were all conducted under humid conditions.

Reply: We express our gratitude to the reviewer for this comment. We take into account the influence of H₂O molecules in characterization tests, such as in situ Raman and DRIFT, as well as in DFT calculations. The inadequate level of details in our description of the characterization tests have resulted in the misinterpretation.

For characterization tests, we have modified the relevant pictures (Fig. 4a, 4f and Supplementary Fig. 34, 38) and descriptions. For DFT calculations, we calculated the free energy for the reaction between the surface atomic oxygen (*O) and H₂O on the MNC catalysts. As shown in Fig. 5f, the CoNC catalyst exhibits the lowest •OH formation barrier (0.62 eV), followed by FeNC (0.85 eV), NiNC (0.96 eV), and MnNC (1.60 eV), which is consistent with the ESR results and the catalytic performance. These findings suggest that the presence of H₂O molecules is crucial in facilitating the catalytic ozonation reaction of the MNC catalysts by interacting with *O, which results in the generation of •OH (2.7 V).

The revised manuscript description is provided below to prevent any potential misunderstandings.

Line 430, “The free energy for the reaction between the *O and H₂O to form •OH was further calculated in Fig. 5f. The CoNC catalyst exhibits the lowest •OH formation barrier (0.62 eV), followed by FeNC (0.85 eV), NiNC (0.96 eV), and MnNC (1.60 eV), which is consistent with the ESR results and the catalytic performance.”

Line 537, “The humidity level was set to 50%.”

Line 547, “Next, the catalyst underwent the adsorption process with 50 ppm of CH₃SH under an N₂ atmosphere for 40 min and then underwent the catalytic ozonation process under an O₃ atmosphere (50 ppm) for 40 min at 50% humidity.”

Figure 4. **Surface chemical reaction studies. a** In situ Raman spectra of the CoNC catalyst in the O₃ atmosphere.

Figure 4. **Surface chemical reaction studies. f** In situ diffuse reflectance infrared Fourier transform spectroscopy (DRIFT) of the catalytic ozonation process over the CoNC.

Figure 5. **Mechanistic insights by DFT calculations. f** Free energy diagrams for the reactions of *O with H₂O molecules over the MNC catalysts.

Supplementary Figure 34. In situ Raman spectra of the MNC catalysts in the O₃ atmosphere (a–d MnNC, FeNC, CoNC, and NiNC).

Supplementary Figure 38. In situ diffuse reflectance infrared Fourier transform (DRIFT) spectroscopy of the catalytic ozonation processes over the MnNC (a), FeNC (b), CoNC (c), and NiNC (d).

Comment 3: *Line 186. It was mentioned that the catalysts exhibit the decreasing O₃ decomposition efficiencies after running for 30 min, which may be owing to the occupation of active sites by intermediates. So, is there a way to restore the activity of catalysts?*

Reply: We express our gratitude to the reviewer for this comment. The surface reaction intermediates of O₃ decomposition on the M₁-N₃C₁ active sites were revealed by in situ Raman. With continuous O₃ flow (Fig. 4a and Supplementary Fig. 34, after N₂ purging), new peaks were observed at 821 and 909 cm⁻¹, which were attributed to the surface peroxide species (*O₂, 1.35 V) and surface atomic oxygen (*O, 2.43 V), respectively.^{1,2} DFT calculations (Fig. 5a–b) revealed that the decrease in O₃ decomposition efficiencies of the MNC catalysts over time was mainly due to the stable *O₂ covering the M₁-N₃C₁ active sites. To restore the catalytic activity, it is necessary to address the desorption of *O₂ on the active sites.³

To regenerate the catalyst, the spent CoNC catalyst was treated for 1 h at 100 °C under an N₂ atmosphere. The results (Supplementary Fig. 19) showed that the activity of CoNC could be fully restored to its original state.

To clarify this result, we have revised the manuscript description as follows.

Line 206, “The decrease in O₃ decomposition activities can be attributed to the occupation of active sites by intermediates, while the O₃ decomposition performance can be restored after 1 h of treatment at 100 °C under the N₂ atmosphere (Supplementary Fig. 19).⁴ Moreover, the metal-dependent performance of O₃ decomposition can be attributed to variances in the interactions between single metal atoms and intermediates, which will be discussed later in the DFT calculations.”

Line 290, “With continuous O₃ flow (Fig. 4a and Supplementary Fig. 34, after N₂ purging), new peaks are observed at 821 and 909 cm⁻¹, which are attributed to the surface peroxide species (*O₂, 1.35 V) and surface atomic oxygen (*O, 2.43 V), respectively, revealing that new intermediates are generated on the M₁-N₃C₁ active sites during O₃ decomposition.^{1,2}”

Line 381, “The Gibbs free energy for each elementary step (eqs 1–3) in O₃ decomposition is calculated to unravel the mechanism for the metal-dependent O₃ decomposition performance (Fig. 5a–b and Supplementary Table 8). The most endothermic steps for the M₁-N₃C₁ active sites are the dissociation of the M-*O₂ complexes with the limiting barrier of MnNC (2.99 eV) > FeNC (2.51 eV) > CoNC (1.49 eV) > NiNC (0.63 eV). Therefore, the different interactions of the single metal atoms and *O₂ lead to the differential dissociation barriers of the M-*O₂ complexes, thus resulting in the metal-dependent O₃ decomposition efficiency. The electronic property of the Ni₁-N₃C₁ active site effectively lowers the dissociation barrier of the *O₂, thus boosting the O₃ decomposition. Moreover, it is confirmed

that the decrease in the O_3 decomposition efficiencies of the MNC catalysts over time is mainly due to the stable $*O_2$ covering the $M_1-N_3C_1$ active sites (Fig. 3a).⁵

Figure 3. O_3 decomposition and catalytic ozonation performance. **a** O_3 dynamic decomposition tests over the NC and MNC catalysts.

Figure 4. **Surface chemical reaction studies.** **a** In situ Raman spectra of the CoNC catalyst in the O_3 atmosphere.

Figure 5. **Mechanistic insights by DFT calculations.** **a** Proposed mechanism of O_3 decomposition over the MNC catalysts. **b** Free energy diagrams along the MNC catalysts catalytic O_3 decomposition pathway. All lengths are given in Å. The blue, silver, brown, and red balls denote M, N, C, and O atoms, respectively.

Supplementary Figure 19. Catalytic O₃ decomposition by continuous use of the CoNC regenerated by annealing.

Supplementary Figure 34. In situ Raman spectra of the MNC catalysts in the O₃ atmosphere (a–d MnNC, FeNC, CoNC, and NiNC).

Supplementary Table 8. Optimize the free energy corresponding to the structure of the resting point in the O₃ decomposition process of the MNC catalysts.

Reaction coordinates	Energy (eV)			
	MnNC	FeNC	CoNC	NiNC
Bare surface + 2O ₃ (g)	0	0	0	0
*O ₃ + O ₃ (g)	-3.09	-2.27	-1.44	-1.00
*O + O ₂ (g) + O ₃ (g)	-3.13	-1.65	-0.50	-0.27
*O ₂ + 2O ₂ (g)	-2.75	-2.26	-1.23	-0.52
Bare surface + 3O ₂ (g)	0.23	0.26	0.26	0.11

References:

1. Bing, J., Hu, C. & Zhang, L. Enhanced mineralization of pharmaceuticals by surface oxidation over mesoporous γ -Ti-Al₂O₃ suspension with ozone. *Appl. Catal. B* **202**, 118-126 (2017).
2. Wang, Y. et al. Occurrence of both hydroxyl radical and surface oxidation pathways in N-doped layered nanocarbons for aqueous catalytic ozonation. *Appl. Catal. B* **254**, 283-291 (2019).
3. Dong, C., Yang, J. J., Xie, L. H., Cui, G., Fang, W. H. & Li, J. R. Catalytic ozone decomposition and adsorptive VOCs removal in bimetallic metal-organic frameworks. *Nat. Commun.* **13**, 4991 (2022).
4. Li, X., He, G., Ma, J., Shao, X., Chen, Y. & He, H. Boosting the dispersity of metallic Ag nanoparticles and ozone decomposition performance of Ag-Mn catalysts via manganese vacancy-dependent metal-support interactions. *Environ. Sci. Technol.* **55**, 16143-16152 (2021).
5. Li, W., Gibbs, G. V. & Oyama, S. T. Mechanism of ozone decomposition on a manganese oxide catalyst. 1. In situ Raman spectroscopy and ab initio molecular orbital calculations. *J. Am. Chem. Soc.* **120**, 9041-9046 (1998).

Comment 4: *The increase of CH₃SH degradation efficiencies of the catalyst at >90 % RH is an interesting phenomenon, so a more detailed explanation is recommended.*

Reply: We express our gratitude to the reviewer for this comment. The increase of CH₃SH degradation efficiencies at > 90% RH can be attributed to the accelerated O₃ decomposition and the generation of •OH with the highest oxidation potential (2.7 V).¹

Our humidity experiment shows that the degradation efficiencies of CoNC, FeNC, NiNC, and MnNC for CH₃SH increase to 100.0%, 94.6%, 89.0%, and 82.8%, respectively, as the relative humidity (RH) > 90% (Supplementary Fig. 26). This phenomenon is quite interesting. Following the suggestion of the reviewer, we conducted O₃ decomposition tests under high humidity conditions. The results show that the MNC catalysts exhibit the best O₃ decomposition performance under high humidity conditions (Supplementary Fig. 27). Therefore, H₂O molecules promote the decomposition of O₃ at the M₁-N₃C₁ active sites. Moreover, the generation of •OH was monitored by electron spin resonance (ESR) tests (Fig. 4c). The acceleration of O₃ decomposition and the subsequent generation of •OH, boosting the degradation efficiency of CH₃SH.

Following the reviewer's comment, the following sections have been added to the revised manuscript.

Line 249, “For practical applications, the catalytic ozonation performance of the MNC catalysts was investigated by varying the various variables, such as humidity, dosages, and contact time. When the relative humidity (RH) > 90% (Supplementary Fig. 26a), the CH₃SH degradation efficiencies of CoNC, FeNC, NiNC, and MnNC increase to 100.0%, 94.6%, 89.0%, and 82.8%, respectively. While, for the dry inlet

gas ($RH < 1\%$), the catalytic activities of the samples become different (Supplementary Fig. 26b). For CoNC, FeNC, NiNC, and MnNC, their CH_3SH degradation efficiencies are 100.0%, 87.9%, 84.0%, and 78.3%, respectively, which subsequently decrease to 95.8%, 74.0%, 70.9%, and 57.8% after 60 min. These results are consistent with the O_3 decomposition performance of the MNC catalysts under both dry and wet conditions. As shown in Supplementary Fig. 27a, under dry conditions, the MNC catalysts exhibit gradual deactivation after 20 min, despite achieving the complete removal of O_3 . The stability of the MNC catalysts exhibits a gradual increase with an elevation in humidity levels (Fig. 3a and Supplementary Fig. 27b). When the humidity exceeds 90% (Supplementary Fig. 27b), the NiNC and CoNC remain active, while the FeNC and MnNC exhibit deactivation after 50 min and 35 min, respectively. These findings suggest that the H_2O molecule not only promotes O_3 decomposition, but also enhances the catalytic ozonation performance of the MNC catalysts by facilitating O_3 decomposition. Notably, the enhanced decomposition of O_3 by H_2O can be attributed to the reaction between H_2O and the intermediates resulting from O_3 decomposition. This will be discussed later in DFT calculations.”

Line 328, “When dimethylpyridine N-oxide (DMPO) is used as a trapping agent in Fig. 4c, the signal of $\cdot\text{OH}$ (2.7 V) is detected, which supports the enhanced CH_3SH degradation performance under high humidity conditions.”

Figure 3. **O_3 decomposition and catalytic ozonation performance.** a O_3 dynamic decomposition tests over the NC and MNC catalysts.

Figure 4. **Surface chemical reaction studies.** Electron spin resonance (ESR) spectra of $\text{DMPO}\cdot\text{OH}$ (c) when the MNC catalysts exposing to O_3 in the dark.

Supplementary Figure 26. Catalytic ozonation for CH_3SH degradation tests over the MNC catalysts under different relative humidity (RH) conditions (**a** $\text{RH} > 90\%$. **b** $\text{RH} < 1\%$).

Supplementary Figure 27. Catalytic O_3 decomposition tests over the MNC catalysts under different relative humidity (RH) conditions (**a** $\text{RH} < 1\%$. **b** $\text{RH} > 90\%$).

References:

- Ren, T., Yin, M., Chen, S., Ouyang, C., Huang, X. & Zhang, X. Single-atom Fe-N_4 sites for catalytic ozonation to selectively induce a nonradical pathway toward wastewater purification. *Environ. Sci. Technol.* **57**, 3623-3633 (2023).

Comment 5: Related tests should be supplemented to verify the stability of the catalysts after the catalytic reaction.

Reply: We express our gratitude to the reviewer for this comment. Following the suggestion of the reviewer, we have supplemented related tests (SEM, AC HAADF-STEM, XRD, and Raman) to verify the stability of the catalysts after the catalytic reaction.

As shown in Supplementary Fig. 32, the CoNC after the catalytic reaction maintains the nanosheet-stacking structure. More importantly, single Co atoms persist throughout the entire range of the catalyst. The result is supported by XRD and Raman results (Supplementary Fig. 33), which indicates the absence of any crystalline metals or metal oxides formed during the catalytic reaction. As evidenced by these results, the catalysts exhibit the great stability throughout the reaction.

To clarify this result, we have revised the manuscript description as follows.

Line 280, “Moreover, the morphology and single Co atoms of the CoNC catalyst (Supplementary Fig. 32–33) remain unchanged after the reaction, confirming the exceptional stability of the CoNC catalyst.”

Supplementary Figure 32. Scanning electron microscope (a SEM) and aberration-corrected high-angle annular dark-field scanning transmission electron microscope (b AC HAADF-STEM) images of the used CoNC catalyst.

Supplementary Figure 33. a Powder X-ray diffraction (XRD) pattern of the used CoNC catalyst. b Raman spectrum of the used CoNC catalyst.

Reviewer 3

General Comment: *In this manuscript, Ma et al. presented some results in synthesis of $M_1-N_3C_1$ -based materials for catalytic ozonation of sulfur-contained compounds and the mechanism. In the work, the authors tried to analyze the active sites for the reaction of O_3 activation and organic compound decomposition. However, the manuscript suffers from fatal deficiencies and the mechanism was not well illustrated and conclusive. The manuscript was not suggested to be accepted by Nat Commun. Other comments are listed below.*

Reply: We express our gratitude to the reviewer for providing constructive comments to enhance the quality of this work. Regarding the fatal deficiencies proposed by the reviewer, we have conducted comprehensive characterization and theoretical analyses of the control nitrogen-doped carbon material to eliminate any potential impacts from the nitrogen-doped carbon matrix in the MNC catalyst. We have identified the $M_1-N_3C_1$ as the active site for O_3 decomposition and catalytic ozonation in the MNC catalyst, rather than the nitrogen-doped carbon matrix. Moreover, we have provided the additional clarification on the reaction mechanisms. We have answered the reviewer's comments point by point and rectified any relevant issues. We would greatly appreciate it if the reviewer could reassess our revised manuscript for potential publication in the journal.

Comment 1: *It is well known that carbon is an active catalyst for ozonation and functional groups on carbon also affect the reaction. However, no investigation was conducted. Thus, it is not clear to derive the major role of M-N-C in catalysis.*

Reply: We express our gratitude to the reviewer for this comment. We concur with the reviewer's comment that carbon is an active catalyst for catalytic ozonation and further acknowledge that the functional groups present on carbon also affect the reaction.¹ The reason for not investigating the nitrogen-doped carbon matrix lies in the fact that all characterization (XRD, SEM, TEM, Raman, and XPS) results indicate that the structural characteristics of the nitrogen-doped carbon matrix in MNC catalysts are identical. We apologize for our oversight, as the manuscript lacks clarity on this point. Moreover, metal dependence was observed in both the O_3 decomposition and catalytic ozonation activity tests of the MNC catalysts. The metal-dependent catalytic activity was confirmed through detailed in situ tests and theoretical calculations. Following the reviewer's comment, we have prepared a metal-free nitrogen-doped carbon material known as NC. We have conducted a series of characterization analyses and activity tests for the NC, comparing them with the MNC catalysts to confirm the key role of the $M_1-N_3C_1$ as the active site.

First, the TEM and SEM images (Supplementary Fig. 5) demonstrate that the nanosheet-stacking structure of the NC is identical to that of the MNC. Moreover, the NC shows the identical XRD (Fig. 2a) characteristic peaks (26° (002, graphite) and 44° (101, graphite)) and Raman spectra (Supplementary Fig. 6) as the MNC,

further indicating their structural similarity. Notably, we investigated the surface functional groups of the NC and MNC using FTIR spectroscopy. As shown in Supplementary Fig. 11, the NC and MNC display two peaks at 1264 and 1535 cm^{-1} , which are assigned to the stretching vibrations of C–N and C=N, respectively.² In addition to the single metal atom, the NC and MNC share identical structural characteristics as analyzed above. Based on this conclusion, we investigated the performance of the NC in both O_3 decomposition and catalytic ozonation reactions. As shown in Fig. 3a–b, the O_3 decomposition and catalytic ozonation activities of the NC are significantly inferior to those of the MNC catalysts, indicating that the $\text{M}_1\text{-N}_3\text{C}_1$ active sites play a crucial role in both reactions. Moreover, the CH_3SH removal efficiency of the NC under air conditions (Supplementary Fig. 22) is also inferior to that of the MNC catalysts. ESR and in situ DRIFTS tests were employed to elucidate the root cause of the unsatisfactory catalytic performance exhibited by the NC. As shown in Fig. 4c–e, the NC exhibits the highest concentration of $\bullet\text{O}_2^-$ and the lowest concentration of $^1\text{O}_2$, while no $\bullet\text{OH}$ is generated. This result is consistent with the correlation between reactive oxygen species and catalytic ozonation performance as presented in our manuscript, whereby high concentrations of $\bullet\text{OH}$ and $^1\text{O}_2$ promote improved catalytic ozonation performance. Moreover, during the catalytic ozonation of the NC, only a few weak characteristic peaks representing CH_3OH (1051 cm^{-1}) and SO_3^{2-} (957 cm^{-1}) are observed in the in situ DRIFTS spectra (Supplementary Fig. 40), indicating the limited performance of the NC in catalytic ozonation. Therefore, it is the $\text{M}_1\text{-N}_3\text{C}_1$ active sites rather than the nitrogen-doped carbon matrix that play a crucial role in both O_3 decomposition and catalytic ozonation reactions. DFT calculations further confirm this conclusion (Supplementary Fig. 41–42), as the adsorption of O_3 on C atoms is much weaker than that on metal atoms (Mn -3.09 eV, Fe -2.27 eV, Co -1.44 eV, Ni -1.00 eV, C -0.88 eV).

To clarify this result, we have revised the manuscript description as follows.

Line 128, “Moreover, the control carbon material (NC) also exhibits the nanosheet-stacking structure (Supplementary Fig. 5), indicating that the introduction of single metal atoms does not alter the morphology of the matrices.”

Line 142, “This is further confirmed by the Fourier transform infrared (FTIR) spectra, as the two peaks at 1264 and 1535 cm^{-1} assigned to the stretching vibrations of C–N and C=N, respectively, are detected (Supplementary Fig. 11).²”

Line 150, “It is worth noting that the matrix structure of the MNC SACs remains consistent, except for variations in the doping type of single metal atoms. This fact is further supported by structural characterization results (Fig. 2a, Supplementary Fig. 6 and 11), which show no discernible differences between the NC and the MNC SACs.”

Line 171, “The overall structural analyses indicate that the MNC SACs have the well-defined $M_1-N_3C_1$ units, which are anchored onto the same nitrogen-doped carbon matrices.”

Line 204, “Notably, the O_3 decomposition efficiencies after 60 min are clearly influenced by the anchored single metal atoms in the order of NiNC (100.0%) > CoNC (99.7%) > FeNC (99.4%) > MnNC (98.6%) > NC (97.4%).”

Line 217, “In contrast, the NC shows the lowest catalytic activity (58.5%), which further decreases to 43.2% after 60 min of reaction. Therefore, it can be inferred that the $M_1-N_3C_1$ active sites play a crucial role in catalytic ozonation reactions.”

Line 235, “By integrating the performance results of the O_3 decomposition and catalytic ozonation, it can be inferred that the $M_1-N_3C_1$ units in the MNC catalysts play a pivotal yet distinct role in these two reactions.”

Line 238, “Interestingly, the degradation efficiencies of CH_3SH by the MNC catalysts in the air (Supplementary Fig. 22) show a metal-dependent profile similar to that observed under the O_3 conditions. The order of CH_3SH degradation efficiencies achieved is CoNC (84.5%) > FeNC (71.0%) > NiNC (41.5%) > MnNC (37.7%) > NC (35.8%). However, the CH_3SH degradation efficiencies decrease to 73.0%, 61.4%, 30.7%, 19.4%, and 0% after 60 min, respectively, revealing the weak oxidation capacity of the MNC catalysts in the air.”

Line 331, “Notably, the signal of $\bullet OH$ shows a metal-dependent profile. Explicitly, the CoNC exhibits the strongest $\bullet OH$ signal followed by the FeNC, and the NiNC, MnNC, and NC show negligible $\bullet OH$ signals. Similar results can be found in Fig. 4d, the 1O_2 concentration decreases in the order of CoNC > FeNC > NiNC > MnNC > NC. Interestingly, the generation of $\bullet O_2^-$ exhibits the exact opposite pattern (Fig. 4e), probably owing to the competitive formation between 1O_2 and $\bullet O_2^-$ (0.8 V, $E_0(^1O_2/\bullet O_2^-)$).³⁹”

Line 363, “Notably, during the catalytic ozonation of the NC, only a few weak characteristic peaks representing CH_3OH (1051 cm^{-1}) and SO_3^{2-} (957 cm^{-1}) are observed (Supplementary Fig. 40), indicating the limited performance of the NC in catalytic ozonation. This finding further emphasizes the pivotal role of the $M_1-N_3C_1$ as the active sites in catalytic ozonation reactions.”

Line 375, “The $M_1-N_3C_1$ units induce the locally polarized M–C bonds to capture O_3 molecules onto the M atoms. Therefore, the M atoms are confirmed as the O_3 adsorption sites (Supplementary Fig. 41–42 and Supplementary Table 7, $M-\ast O_3$, E_{ads} (Mn -3.09 eV, Fe -2.27 eV, Co -1.44 eV, Ni -1.00 eV)).”

Line 480, “The nitrogen-doped carbon material, denoted as NC, was synthesized without the addition of any metal precursors while keeping the other processing parameters constant.”

Figure 2. **Atomic structure analysis.** a Powder X-ray diffraction (XRD) patterns of the NC and MNC catalysts.

Figure 3. **O₃ decomposition and catalytic ozonation performance.** a O₃ dynamic decomposition tests over the NC and MNC catalysts. b Catalytic ozonation for CH₃SH degradation tests over the NC and MNC catalysts.

Figure 4. **Surface chemical reaction studies.** Electron spin resonance (ESR) spectra of DMPO·OH (c), ¹O₂ (d), and DMPO·O₂⁻ (e) when the MNC catalysts

exposing to O₃ in the dark. **f** In situ diffuse reflectance infrared Fourier transform spectroscopy (DRIFT) of the catalytic ozonation process over the CoNC.

Supplementary Figure 5. Scanning electron microscope (**a** SEM) and transmission electron microscope (**b** TEM) images of the NC.

Supplementary Figure 6. Raman spectra of the NC and MNC catalysts.

Supplementary Figure 11. Fourier transform infrared (FTIR) spectra of the NC and MNC catalysts.

Supplementary Figure 22. CH₃SH dynamic degradation tests over the NC and MNC catalysts in the air.

Supplementary Figure 40. **a** In situ diffuse reflectance infrared Fourier transform (DRIFT) spectroscopy of the NC in the CH₃SH/N₂ atmosphere. **b** In situ DRIFT spectroscopy of the catalytic ozonation processes over the NC.

Supplementary Figure 41. The optimized adsorption of O₃ molecule on the MNC coordination structures (**a–d** CoNC, FeNC, NiNC, and MnNC). All lengths are given in Å. The purple, yellow, blue, silver gray, brown, red, and silver balls denote Mn, Fe, Co, Ni, C, O, and N atoms, respectively.

Supplementary Figure 42. The optimized adsorption of O₃ molecule on C atoms. All lengths are given in Å. The blue, brown, red, and silver balls denote Co, C, O, and N atoms, respectively.

Supplementary Table 7. Optimize the adsorption energy corresponding to the structure of the resting point in the molecular adsorption process of the MNC catalysts.

Catalysts	Adsorption energy (eV)					
	O ₃	CH ₃ SH	O ₂	H ₂ O	CO ₂	SO ₄ ²⁻
CoNC	-1.44	-0.45	-1.49	-0.32	-0.21	-0.13
FeNC	-2.27	-1.22	-2.51	-0.37	-0.20	-0.15
NiNC	-1.00	-0.35	-0.63	-0.24	-0.18	-0.15
MnNC	-3.09	-1.39	-2.99	-0.62	-0.59	-0.87
NC	-0.88	/	/	/	/	/

References:

1. Wang, Y., Duan, X., Xie, Y., Sun, H. & Wang, S. Nanocarbon-based catalytic ozonation for aqueous oxidation: engineering defects for active sites and tunable reaction pathways. *ACS Catal.* **10**, 13383-13414 (2020).
2. Cai, Y. et al. Insights on forming N,O-coordinated Cu single-atom catalysts for electrochemical reduction CO₂ to methane. *Nat. Commun.* **12**, 586 (2021).
3. Nosaka, Y. & Nosaka, A. Y. Generation and detection of reactive oxygen species in photocatalysis. *Chem. Rev.* **117**, 11302-11336 (2017).

Comment 2: *The authors should test the control carbon sample in catalysis.*

Reply: We express our gratitude to the reviewer for this comment. Following the suggestion of the reviewer, we have performed O₃ decomposition and catalytic ozonation tests on the control carbon samples (NC). As shown in Fig. 3a–b, the O₃ decomposition and catalytic ozonation activities of the NC are significantly inferior to those of the MNC catalysts, indicating that the M₁-N₃C₁ active sites play a crucial role in both reactions. Moreover, the CH₃SH removal efficiency of the NC under air conditions (Supplementary Fig. 22) is also inferior to that of the MNC catalysts.

To clarify this result, we have revised the manuscript description as follows.

Line 204, “Notably, the O₃ decomposition efficiencies after 60 min are clearly influenced by the anchored single metal atoms in the order of NiNC (100.0%) > CoNC (99.7%) > FeNC (99.4%) > MnNC (98.6%) > NC (97.4%).”

Line 217, “In contrast, the NC shows the lowest catalytic activity (58.5%), which further decreases to 43.2% after 60 min of reaction. Therefore, it can be inferred that the M₁-N₃C₁ active sites play a crucial role in catalytic ozonation reactions.”

Line 235, “By integrating the performance results of the O₃ decomposition and catalytic ozonation, it can be inferred that the M₁-N₃C₁ units in the MNC catalysts play a pivotal yet distinct role in these two reactions.”

Line 238, “Interestingly, the degradation efficiencies of CH₃SH by the MNC catalysts in the air (Supplementary Fig. 22) show a metal-dependent profile similar to that observed under the O₃ conditions. The order of CH₃SH degradation efficiencies achieved is CoNC (84.5%) > FeNC (71.0%) > NiNC (41.5%) > MnNC (37.7%) > NC (35.8%). However, the CH₃SH degradation efficiencies decrease to 73.0%, 61.4%, 30.7%, 19.4%, and 0% after 60 min, respectively, revealing the weak oxidation capacity of the MNC catalysts in the air.”

Figure 3. O₃ decomposition and catalytic ozonation performance. **a** O₃ dynamic decomposition tests over the NC and MNC catalysts. **b** Catalytic ozonation for CH₃SH degradation tests over the NC and MNC catalysts.

Supplementary Figure 22. CH₃SH dynamic degradation tests over the NC and MNC catalysts in the air.

Comment 3: *The authors did not suggest the detailed metal chemical state in the samples by XPS.*

Reply: We express our gratitude to the reviewer for this comment. Following the suggestion of the reviewer, we have conducted comprehensive X-ray photoelectron spectroscopy (XPS) analyses to determine the chemical state of metals in the samples.

To clarify this result, we have revised the manuscript description as follows.

Line 136, “The chemical elements and electronic states of the MNC investigated by X-ray photoelectron spectroscopy (XPS, Supplementary Fig. 7–10) show that the metal species only present in their oxidation states (Mn²⁺ 641.53 eV, Fe²⁺ 709.96 eV, Co²⁺ 780.62 eV, Ni²⁺ 854.72 eV) rather than metallic states.¹⁻⁴”

References:

1. Ye, J. et al. Tumor response and NIR-II photonic thermal co-enhanced catalytic therapy based on single-atom manganese nanozyme. *Adv. Funct. Mater.* **32**, 2206157 (2022).
2. Sun, X. et al. Phosphorus induced electron localization of single iron sites for boosted CO₂ electroreduction reaction. *Angew. Chem. Int. Ed. Engl.* **60**, 23614-23618 (2021).
3. Ge, K. et al. Facile synthesis of two-dimensional iron/cobalt metal-organic framework for efficient oxygen evolution electrocatalysis. *Angew. Chem. Int. Ed. Engl.* **60**, 12097-12102 (2021).
4. Wang, X. et al. Dynamic activation of adsorbed intermediates via axial traction for the promoted electrochemical CO₂ reduction. *Angew. Chem. Int. Ed. Engl.* **60**, 4192-4198 (2021).

Comment 4: *It is also well known that carbon-based catalytic ozonation will produce different reactive species and the authors did not investigate their reactions with sulfur compounds. Thus, it is difficult to link the active site to organic decomposition.*

Reply: We express our gratitude to the reviewer for this comment. We concur with the reviewer's comment that carbon-based materials will generate different reactive species (*O, *O₂, •OH, •O₂⁻, and ¹O₂) in an O₃ atmosphere.¹ We have confirmed the generation of these reactive oxygen species (*O, *O₂, •OH, •O₂⁻, and ¹O₂) during the catalytic ozonation process using in-situ Raman (Fig. 4a and Supplementary Fig. 34) and ESR (Fig. 4c–e) tests on the MNC catalysts. Given the uncertainty inherent in quenching experiments, it is exceedingly challenging to discern the contribution of individual reactive oxygen species to CH₃SH removal.¹ Therefore, the understanding of the structure-activity relationship still remains obscure in catalytic ozonation, which obscures the origin of the activity and limits the optimal design of the catalysts.

Recent studies have proposed that in catalytic ozonation reactions, the $*O$ and $*O_2$ not only directly participate in oxidation reactions but also regulate the generation of other reactive oxygen species, such as $\bullet OH$, 1O_2 , and $\bullet O_2^-$, by reacting with H_2O , thus indirectly affecting catalytic activity.¹ Therefore, it is reasonable to consider that the $*O/*O_2$ can be the potential descriptors to directly reflect the structure-activity relationships in catalytic ozonation.

To reveal the correlation between $*O/*O_2$ and catalytic performance, we first studied the reactivity of $*O/*O_2$ by amperometric i-t curve (i-t) tests. As shown in Fig. 4b, the sequential addition of saturated O_3 solution and sodium thiomethoxide (CH_3NaS) solution shows a significant current impulse, indicating the formation of $M-*O$ and/or $M-*O_2$ complexes and the subsequent oxidation of CH_3NaS .² The reactivity of the complexes ($M-*O/*O_2$, based on the degree of the current impulse in Supplementary Fig. 36) relies on the variety of single metal atoms following the order of $CoNC > FeNC > NiNC > MnNC$, which is consistent with the catalytic ozonation performance (Fig. 3b). This result confirms the direct involvement of $M-*O/*O_2$ in the catalytic ozonation reactions and highlights the direct effects of $M-*O/*O_2$ reactivity on catalytic ozonation performance. Moreover, the presence of other reactive oxygen species was studied by the electron spin resonance (ESR) tests. It is found that the concentration order of the generated $\bullet OH$ and 1O_2 is consistent with the reactivity of $M-*O/*O_2$ and catalytic ozonation performance of the MNC catalysts, indicating that the $\bullet OH$ and 1O_2 are regulated by the reactivity of $M-*O/*O_2$ and play a crucial role in the CH_3SH oxidation. Therefore, we propose that the functional $M_1-N_3C_1$ active sites capture and dissociate O_3 molecules, forming the $M-*O/*O_2$ complexes and subsequently governing the generation of $\bullet OH/^1O_2/\bullet O_2^-$, thus achieving the efficient degradation of CH_3SH . Furthermore, we suggest that the reactivity of $*O/*O_2$, which has a direct or indirect impact on catalytic activity, can serve as a potential descriptor to reveal the structure-activity relationship in catalytic ozonation. The density functional theory (DFT) calculations further confirmed these findings. The Gibbs free energy was calculated for both the oxidation reactions in which the $*O$ directly participates and the reactions in which the $*O$ mediates the formation of other reactive oxygen species. The results (Fig. 5e-f) confirm that the $*O$ not only directly participates in oxidation reactions but also regulates the generation of other reactive oxygen species, thus governing catalytic activity.

To clarify this result, we have revised the manuscript description as follows.

Line 64, “Recent studies have proposed that in catalytic ozonation reactions, the $*O$ and $*O_2$ not only directly participate in oxidation reactions but also regulate the generation of other reactive oxygen species, such as hydroxyl radicals ($\bullet OH$), singlet oxygen (1O_2), and superoxide radicals ($\bullet O_2^-$), by reacting with H_2O , thus indirectly affecting catalytic activity.^{3,4}”

Line 315, “The reactivity of $^*O/^*O_2$, directly determined by the $M_1-N_3C_1$ active sites, was investigated ... can serve as a potential descriptor to reveal the structure-activity relationship in catalytic ozonation.”

Line 421, “To visually demonstrate the role of *O as a descriptor, the Gibbs free energy was calculated for both ... thus lowering the reaction barriers for the oxidation of $C_2H_6O_2S$ and maximizing the generation of reactive oxygen species ($\bullet OH$ and 1O_2) with high oxidation potentials.”

Figure 3. O_3 decomposition and catalytic ozonation performance. **b** Catalytic ozonation for CH_3SH degradation tests over the NC and MNC catalysts.

Supplementary Figure 34. In situ Raman spectra of the MNC catalysts in the O_3 atmosphere (**a–d** MnNC, FeNC, CoNC, and NiNC).

Figure 4. **Surface chemical reaction studies.** **a** In situ Raman spectra of the CoNC catalyst in the O_3 atmosphere. **b** The amperometric i-t curves on the MNC catalysts. Electron spin resonance (ESR) spectra of $DMPO-\bullet OH$ (**c**), 1O_2 (**d**), and $DMPO-\bullet O_2^-$ (**e**) when the MNC catalysts exposing to O_3 in the dark. **f** In situ diffuse reflectance infrared Fourier transform spectroscopy (DRIFT) of the catalytic ozonation process over the CoNC.

Figure 5. **Mechanistic insights by DFT calculations.** **e** Free energy diagrams for the reactions of *O with $C_2H_6O_2S$ molecules over the MNC catalysts. **f** Free energy diagrams for the reactions of *O with H_2O molecules over the MNC catalysts.

Supplementary Figure 36. The corresponding current variation of the chronoamperometry curves on the MNC catalysts.

References:

1. Wang, Y., Duan, X., Xie, Y., Sun, H. & Wang, S. Nanocarbon-based catalytic ozonation for aqueous oxidation: engineering defects for active sites and tunable reaction pathways. *ACS Catal.* **10**, 13383-13414 (2020).
2. Wang, Y. et al. Occurrence of both hydroxyl radical and surface oxidation pathways in N-doped layered nanocarbons for aqueous catalytic ozonation. *Appl. Catal. B* **254**, 283-291 (2019).
3. Ren, T., Yin, M., Chen, S., Ouyang, C., Huang, X. & Zhang, X. Single-atom Fe-N₄ sites for catalytic ozonation to selectively induce a nonradical pathway toward wastewater purification. *Environ. Sci. Technol.* **57**, 3623-3633 (2023).
4. Yu, G. et al. Insights into the mechanism of ozone activation and singlet oxygen generation on N-doped defective nanocarbons: a DFT and machine learning study. *Environ. Sci. Technol.* **56**, 7853-7863 (2022).

Comment 5: In N_2 atmosphere, it was suggested that mild oxidation of CH_3SH by the $M-O_2^*$ complexes. How will the complexes be generated?

Reply: We express our gratitude to the reviewer for this comment. The generation of the $M-^*O_2$ complexes can be attributed to the exposure of the MNC catalysts to air prior to in situ diffuse reflectance infrared Fourier transform spectroscopy (DRIFT) tests. The Raman results confirm this result. As shown in Supplementary Fig. 35, the characteristic peak for the *O_2 contribution (821 cm^{-1}) can be found in the Raman spectra of the catalysts performed in the air.¹ Based on the results of in situ DRIFT, when the mixture of N_2 and CH_3SH is introduced (Supplementary Fig. 37, after N_2 purging), two weak bands at 2941 and 806 cm^{-1} are observed, corresponding to the antisymmetric stretching mode of CH_3 and the stretching mode of $S-O$ bonds, respectively.² This result demonstrates the mild oxidation of CH_3SH . The mild oxidation of CH_3SH can be attributed to the $M-^*O_2$ complexes.

To clarify this result, we have revised the manuscript description as follows.

Line 307, “Notably, the characteristic peak for *O_2 contribution can be found in the Raman spectra of the catalysts (Supplementary Fig. 35) performed in the air.¹ Therefore, the mild oxidation of CH_3SH in the air (Supplementary Fig. 22–24) can be attributed to the presence of *O_2 (1.35 V).¹”

Line 354, “This result demonstrates that the stable $M-^*O_2$ complexes formed during air exposure prior to the tests exert a mild oxidizing effect on CH_3SH .”

Supplementary Figure 22. CH_3SH dynamic degradation tests over the NC and MNC catalysts in the air.

Supplementary Figure 23. The concentrations of CH_3SH and typical intermediates in the outlet gases of the MNC catalysts after the CH_3SH dynamic degradation tests in the air for 60 min determined by proton transfer reaction time-of-flight mass spectrometry (PTR-TOF-MS) (a–d MnNC, FeNC, CoNC, and NiNC).

Supplementary Figure 24. The concentrations of CH₃SH and typical intermediates in the outlet gases of the MnNC catalysts after the CH₃SH dynamic degradation tests in the air for 60 min determined by proton transfer reaction time-of-flight mass spectrometry (PTR-TOF-MS).

Supplementary Figure 35. Raman spectra of the MnNC catalysts in the air.

Supplementary Figure 37. In situ diffuse reflectance infrared Fourier transform (DRIFT) spectroscopy of the MnNC (a), FeNC (b), CoNC (c), and NiNC (d) in the CH₃SH/N₂ atmosphere.

References:

1. Huang, Y. et al. Enhanced catalytic ozonation for eliminating CH₃SH via graphene-supported positively charged atomic Pt undergoing Pt²⁺/Pt⁴⁺ redox cycle. *Environ. Sci. Technol.* **55**, 16723-16734 (2021).
2. Qi, Z., Chen, L., Zhang, S., Su, J. & Somorjai, G. A. Mechanism of methanol decomposition over single-site Pt₁/CeO₂ catalyst: a DRIFTS study. *J. Am. Chem. Soc.* **143**, 60-64 (2021).

Comment 6: *In DFT calculations, E_{ads} of different metals suggested the strongest adsorption of O₃ on Mn, which indicates the favored thermodynamics. However, Mn-based material was not good in catalysis. Other calculations seem not to agree with the tests of different metals.*

Reply: We express our gratitude to the reviewer for this comment. We concur with the reviewer's comment that E_{ads} of different metals suggest the strongest adsorption of O₃ on Mn, indicating the favorable thermodynamics. However, the thermodynamic favorability does not necessarily correlate with excellent catalytic activity. The catalytic activity is also controlled by kinetic favorability.¹ For example, Chen et al. proposed that the ΔG(*HOCO)-ΔG(*CO) free energy difference serves as a more suitable descriptor for predicting the performance of CO₂ reduction reaction, rather than relying on CO₂ adsorption energy alone.² Xie et al. determined that the O₃ decomposition performance is characterized by the activation potential required for ³O₂ generation.³ Therefore, catalytic activity is determined by the combined thermodynamic and kinetic favorability.⁴ Predicting catalytic activity solely based on reactant adsorption energy, which represents thermodynamic favorability, is both unscientific and unreasonable. This finding is also supported by our study. Combined with comprehensive catalytic testing, in situ characterization techniques, and density functional theory (DFT) calculations, we propose that the reactivity of surface atomic oxygen (*O) and the dissociation barrier of surface peroxide species (*O₂) can serve as descriptors for catalytic ozonation and O₃ decomposition, respectively.

Our calculations are in complete agreement with the results of our catalytic tests. The Gibbs free energy for each elementary step in O₃ decomposition is calculated to unravel the mechanisms for the metal-dependent O₃ decomposition performance (Fig. 5a–b and Supplementary Table 8). The most endothermic steps for the M₁-N₃C₁ active sites are the dissociation of the M-*O₂ complexes with the limiting barrier of MnNC (2.99 eV) > FeNC (2.51 eV) > CoNC (1.49 eV) > NiNC (0.63 eV). Therefore, the superior O₃ decomposition performance of the NiNC is attributed to its lowest dissociation potential for *O₂, which is consistent with the O₃ decomposition performance shown in Fig. 3a. For the catalytic ozonation, we have

calculated the free energy changes for the reactions of $\cdot\text{O}$ with $\text{C}_2\text{H}_6\text{O}_2\text{S}$ (Fig. 5e), as well as the free energy changes for the reactions of $\cdot\text{O}$ with H_2O to generate $\cdot\text{OH}$ (Fig. 5f). The findings are in line with the catalytic ozonation performance (Fig. 3b) and ESR characterization results (Fig. 4c). Therefore, we propose that the reactivity of surface atomic oxygen ($\cdot\text{O}$) and the dissociation barrier of surface peroxide species ($\cdot\text{O}_2$) can serve as descriptors for catalytic ozonation and O_3 decomposition, respectively.

To clarify this result, we have revised the manuscript description as follows.

Line 381, “The Gibbs free energy for each elementary step (eqs 1–3) in O_3 decomposition is calculated ... effectively lowers the dissociation barrier of the $\cdot\text{O}_2$, thus boosting the O_3 decomposition.”

Line 426, “As shown in Fig. 5e, the free energy for the reaction of $\cdot\text{O}$ with $\text{C}_2\text{H}_6\text{O}_2\text{S}$ decreases ... lowering the reaction barriers for the oxidation of $\text{C}_2\text{H}_6\text{O}_2\text{S}$ and maximizing the generation of reactive oxygen species ($\cdot\text{OH}$ and $^1\text{O}_2$) with high oxidation potentials.”

Figure 3. O_3 decomposition and catalytic ozonation performance. **a** O_3 dynamic decomposition tests over the NC and MNC catalysts. **b** Catalytic ozonation for CH_3SH degradation tests over the NC and MNC catalysts.

Figure 4. **Surface chemical reaction studies.** Electron spin resonance (ESR) spectra of $\text{DMPO}\cdot\text{OH}$ (c) when the MNC catalysts exposing to O_3 in the dark.

Figure 5. **Mechanistic insights by DFT calculations.** **a** Proposed mechanism of O_3 decomposition over the MNC catalysts. **b** Free energy diagrams along the MNC catalysts catalytic O_3 decomposition pathway. All lengths are given in Å. The blue, silver, brown, and red balls denote M, N, C, and O atoms, respectively.

Figure 5. **Mechanistic insights by DFT calculations.** **e** Free energy diagrams for the reactions of $*O$ with $C_2H_6O_2S$ molecules over the MNC catalysts. **f** Free energy diagrams for the reactions of $*O$ with H_2O molecules over the MNC catalysts.

Supplementary Table 8. Optimize the free energy corresponding to the structure of the resting point in the O_3 decomposition process of the MNC catalysts.

Reaction coordinates	Energy (eV)			
	MnNC	FeNC	CoNC	NiNC
Bare surface + $2O_3$ (g)	0	0	0	0
$*O_3 + O_3$ (g)	-3.09	-2.27	-1.44	-1.00
$*O + O_2$ (g) + O_3 (g)	-3.13	-1.65	-0.50	-0.27
$*O_2 + 2O_2$ (g)	-2.75	-2.26	-1.23	-0.52
Bare surface + $3O_2$ (g)	0.23	0.26	0.26	0.11

References:

1. Riscoe, A. R. et al. Transition state and product diffusion control by polymer-nanocrystal hybrid catalysts. *Nat. Catal.* **2**, 852-863 (2019).
2. Chang, Q. et al. Metal-coordinated phthalocyanines as platform molecules for understanding isolated metal sites in the electrochemical reduction of CO_2 . *J. Am. Chem. Soc.* **144**, 16131-16138 (2022).

3. Yu, G. et al. Insights into the mechanism of ozone activation and singlet oxygen generation on N-doped defective nanocarbons: a DFT and machine learning study. *Environ. Sci. Technol.* **56**, 7853-7863 (2022).
4. Kim, J. et al. Tailoring binding abilities by incorporating oxophilic transition metals on 3D nanostructured Ni arrays for accelerated alkaline hydrogen evolution reaction. *J. Am. Chem. Soc.* **143**, 1399-1408 (2021).

Comment 7: *The conclusion on O^* and $*O_2$ as descriptors for catalytic ozonation and O_3 decomposition was not convincing as the authors did not well consider other species involved in catalytic ozonation.*

Reply: We express our gratitude to the reviewer for this comment. We conducted a comprehensive analysis of the reactive oxygen species (surface atomic oxygen ($*O$), surface peroxide species ($*O_2$), hydroxyl radicals ($\bullet OH$), singlet oxygen (1O_2), and superoxide radicals ($\bullet O_2^-$)) involved in catalytic ozonation.

Recent studies have proposed that in catalytic ozonation reactions, the $*O$ and $*O_2$ not only directly participate in oxidation reactions but also regulate the generation of other reactive oxygen species, such as $\bullet OH$, 1O_2 , and $\bullet O_2^-$, by reacting with H_2O , thus indirectly affecting catalytic activity.¹ Therefore, it is reasonable to consider that $*O/*O_2$ can be the potential descriptors to directly reflect the structure-activity relationships in catalytic ozonation.

To reveal the correlation between $*O/*O_2$ and catalytic performance, we first studied the reactivity of $*O/*O_2$ by amperometric i-t curve (i-t) tests. As shown in Fig. 4b, the sequential addition of saturated O_3 solution and sodium thiomethoxide (CH_3NaS) solution shows a significant current impulse, indicating the formation of $M-*O$ and/or $M-*O_2$ complexes and the subsequent oxidation of CH_3NaS .² The reactivity of the complexes ($M-*O/*O_2$, based on the degree of the current impulse in Supplementary Fig. 36) relies on the variety of single metal atoms following the order of $CoNC > FeNC > NiNC > MnNC$, which is consistent with the catalytic ozonation performance (Fig. 3b). This result confirms the direct involvement of $M-*O/*O_2$ in the catalytic ozonation reactions and highlights the direct effects of $M-*O/*O_2$ reactivity on catalytic ozonation performance. Moreover, the presence of other reactive oxygen species was studied by the electron spin resonance (ESR) tests. It is found that the concentration order of the generated $\bullet OH$ and 1O_2 is consistent with the reactivity of $M-*O/*O_2$ and catalytic ozonation performance of the MNC catalysts, indicating that the $\bullet OH$ and 1O_2 are regulated by the reactivity of $M-*O/*O_2$ and play a crucial role in the CH_3SH oxidation. Therefore, we propose that the functional $M_1-N_3C_1$ active sites capture and dissociate O_3 molecules, forming the $M-*O/*O_2$ complexes and subsequently governing the generation of $\bullet OH/^1O_2/\bullet O_2^-$, thus achieving the efficient degradation of CH_3SH . Furthermore, we suggest that the reactivity of $*O/*O_2$, which has a direct or indirect impact on catalytic activity, can serve as a potential descriptor to reveal the structure-activity relationship in catalytic ozonation. The density functional theory

(DFT) calculations further confirmed these findings. The Gibbs free energy was calculated for both the oxidation reactions in which the *O directly participates and the reactions in which the *O mediates the formation of other reactive oxygen species. The results (Fig. 5e–f) confirm that the *O not only directly participates in oxidation reactions but also regulates the generation of other reactive oxygen species, thus governing catalytic activity.

To clarify this result, we have revised the manuscript description as follows.

Line 64, “Recent studies have proposed that in catalytic ozonation reactions, the *O and *O₂ not only directly participate in oxidation reactions but also regulate the generation of other reactive oxygen species, such as hydroxyl radicals (•OH), singlet oxygen (¹O₂), and superoxide radicals (•O₂⁻), by reacting with H₂O, thus indirectly affecting catalytic activity.^{3,4}”

Line 315, “The reactivity of *O/*O₂, directly determined by the M₁-N₃C₁ active sites, was investigated ... can serve as a potential descriptor to reveal the structure-activity relationship in catalytic ozonation.”

Line 421, “To visually demonstrate the role of *O as a descriptor, the Gibbs free energy was ...the oxidation of C₂H₆O₂S and maximizing the generation of reactive oxygen species (•OH and ¹O₂) with high oxidation potentials.”

Figure 3. O₃ decomposition and catalytic ozonation performance. **b** Catalytic ozonation for CH₃SH degradation tests over the NC and MNC catalysts.

Figure 5. **Mechanistic insights by DFT calculations.** **e** Free energy diagrams for the reactions of $\cdot\text{O}$ with $\text{C}_2\text{H}_6\text{O}_2\text{S}$ molecules over the MNC catalysts. **f** Free energy diagrams for the reactions of $\cdot\text{O}$ with H_2O molecules over the MNC catalysts.

Figure 4. **Surface chemical reaction studies.** **a** In situ Raman spectra of the CoNC catalyst in the O_3 atmosphere. **b** The amperometric i-t curves on the MNC catalysts. Electron spin resonance (ESR) spectra of $\text{DMPO}\cdot\text{OH}$ (**c**), $^1\text{O}_2$ (**d**), and $\text{DMPO}\cdot\text{O}_2^-$ (**e**) when the MNC catalysts exposing to O_3 in the dark. **f** In situ diffuse reflectance infrared Fourier transform spectroscopy (DRIFT) of the catalytic ozonation process over the CoNC.

Supplementary Figure 34. In situ Raman spectra of the MNC catalysts in the O_3 atmosphere (a–d MnNC, FeNC, CoNC, and NiNC).

Supplementary Figure 36. The corresponding current variation of the chronoamperometry curves on the MNC catalysts.

References:

1. Wang, Y., Duan, X., Xie, Y., Sun, H. & Wang, S. Nanocarbon-based catalytic ozonation for aqueous oxidation: engineering defects for active sites and tunable reaction pathways. *ACS Catal.* **10**, 13383-13414 (2020).
2. Wang, Y. et al. Occurrence of both hydroxyl radical and surface oxidation pathways in N-doped layered nanocarbons for aqueous catalytic ozonation. *Appl. Catal. B* **254**, 283-291 (2019).
3. Ren, T., Yin, M., Chen, S., Ouyang, C., Huang, X. & Zhang, X. Single-atom Fe-N₄ sites for catalytic ozonation to selectively induce a nonradical pathway toward wastewater purification. *Environ. Sci. Technol.* **57**, 3623-3633 (2023).
4. Yu, G. et al. Insights into the mechanism of ozone activation and singlet oxygen generation on N-doped defective nanocarbons: a DFT and machine learning study. *Environ. Sci. Technol.* **56**, 7853-7863 (2022).

REVIEWERS' COMMENTS

Reviewer #1 (Remarks to the Author):

After careful evaluation of revised manuscript, I believe that the manuscript can be suitable for publication in journal "Nature Communications" in presented form. All necessary additions and clarifications are made in the revised manuscript.

Manuscript ID: NCOMMS-23-17636A

Title: Catalytic ozonation mechanism over $M_1-N_3C_1$ active sites

Authors: Dingren Ma, Qiyu Lian, Yexing Zhang, Yajing Huang, Xinyi Guan, Qiwen Liang, Chun He, Dehua Xia, Shengwei Liu, and Jianguo Yu

General note: The comments from reviewers are presented in black *italic* font style. Responses to each comment are formatted with indentation and displayed in regular style. This document contains the responses to the comments from the reviewer #1.

Reviewer #1

General Comment: *After careful evaluation of revised manuscript, I believe that the manuscript can be suitable for publication in journal “Nature Communications” in presented form. All necessary additions and clarifications are made in the revised manuscript.*

Reply: We express our gratitude to the reviewer for acknowledging the research value and providing constructive comments to enhance the quality of this work.